# High-resolution air quality maps for Bucharest using Mixed-Effects Modeling Framework

Camelia Talianu[1,2,*], Jeni Vasilescu[1,*], Doina Nicolae[1,*], Alexandru Ilie[1,3], Andrei Dandocsi[1,4], Anca Nemuc[1], and Livio Belegante[1]

[1]National Institute of Research and Development for Optoelectronics-INOE 2000, Str. Atomistilor 409, Măgurele, 077125, Ilfov, Romania
[2]Institute of Meteorology and Climatology, Department of Water, Atmosphere and Environment, University of Natural Resources and Life Sciences, Gregor-Mendel Street 33, Vienna, A-1180, Vienna, Austria
[3]Faculty of Geography, University of Bucharest, Bulevardul Nicolae Bălcescu 1, Bucharest, 010041, Bucharest, Romania
[4]UNST Politehnica of Bucharest, Bulevardul Iuliu Maniu 1-3, Bucharest, 061071, Bucharest, Romania
[*]These authors contributed equally to this work.

**Correspondence:** Camelia Talianu (camelia@inoe.ro), Jeni Vasilescu (jeni@inoe.ro), and Doina Nicolae (nnicol@inoe.ro)

**Abstract.** High-resolution mapping of pollutants based on mobile observations facilitates deep understanding of air pollutants distribution within a city. This approach fosters science-based decisions to improve air quality, by adding up to the existing but not optimally distributed permanent monitoring stations. In this study, we developed high-resolution concentration maps of nitrogen dioxide ($NO_2$), particulate matter ($PM_{10}$) and ultrafine particles (UFP) for Bucharest, Romania, to evaluate the spatial variation of pollutants across the city during the warm and the cold seasons. Maps were generated using a mixed-effect method applied to a Land-use Regression (LUR) model. The approach relies on multiple land-use and traffic predictor variables, and assimilation of data collected by mobile measurements over 30 days in the periods May–July 2022 and January–February 2023. Cross-validation was done against in-situ data extracted from the same collection, while validation was organized by comparison with standard measurements at fixed reference sites. Our study shows that this combined method has a good performance for all pollutants ($R^2 > 0.65$), the highest performance being observed for the cold season. $PM_{10}$ concentration maps indicate multiple sources of particles during the warm season, the most important source being traffic. During the cold season $PM_{10}$ concentration maps show a more uniform distribution of sources in Bucharest. The city's principal roads, particularly the Bucharest ring road, are also highlighted in the $NO_2$ maps, with higher gradient during the warm period.

*Copyright statement.* TEXT

## 1 Introduction

The atmosphere is an essential element for the environment and life forms on Earth. Therefore, any change in natural composition of the atmosphere due to the presence of one or more pollutants in the atmosphere, such as gases or aerosols released directly into the atmosphere from natural or anthropogenic sources, can dramatically influence the Earth's climate and bio-

sphere, human life and health, and economic activities (Nemuc et al., 2013; Kokkalis et al., 2016; Ilie et al., 2023). Short term exposure to very high concentrations can also be a significant risk factor to human health in addition to prolonged exposure at lower concentrations. One major concern nowadays is the air quality in urban areas due to its significant health risks determined by prolonged population exposure to gaseous pollutants like nitrogen dioxide ($NO_2$), as well as to particulate matter ($PM_{2.5}$ and $PM_{10}$) (Brunekreef and Holgate, 2002; Bernstein et al., 2004; Almetwally et al., 2020). Numerous epidemiological studies related short- and long-term $PM_{10}$ and $NO_2$ exposure with mortality and morbidity. Short-term exposure to high concentrations of pollutants can be related to both minor discomfort, such as irritation of the eyes, respiratory tract, or skin, and serious conditions, such as asthma, pneumonia, bronchitis, chronic obstructive pulmonary disease and heart problems (Rongqi Abbie et al., 2022; Hasegawa et al., 2023). Furthermore, years of continuous exposure to PM were shown to be associated with both newborn mortality and cardiovascular disorders. A $PM_{2.5}$ concentration increase with 10 $\mu g/m^3$ was associated with a increase of 0.67% – 1.04% (Hamanaka and Mutlu, 2018) in all-cause mortality, 0.52% in cardiovascular hospital admissions and 1.74% increase in respiratory admissions (Hasegawa et al., 2023). While, a $PM_{10}$ concentration increase with 10 $\mu g/m^3$ was associated with a 43% increase of fatal coronary heart disease (Hamanaka and Mutlu, 2018) and 39.31% of deaths from cardiovascular diseases from short-term exposure (Seihei et al., 2024). A smaller impact is foreseen in the case of short-term exposure to $NO_2$ concentration, when an 10 ppb increase of concentration was associated with 0.19% increase in all-cause mortality in US (Hamanaka and Mutlu, 2018). Despite its critical impact (West et al., 2016), air pollution information in urban areas is not always available, or not at an appropriate spatial resolution, hindering effective air quality management efforts. High-resolution air quality maps are pivotal for environmental stewardship and public awareness by filling the gaps in our understanding of urban air quality. These maps can help to identify pollution hotspots, offering new opportunities for pollution mitigation strategies and influencing both policy and individual behavior (Apte et al., 2017; Schmitz et al., 2019).

Mapping pollutant concentrations in urban areas requires fine-scale spatial interpolation of data collected at air quality monitoring stations, taking into account known emission sources and sinks to estimate the actual distribution of pollutants at ambient surface level. Moreover, changes in the composition of the atmosphere caused by urban agglomeration is highly variable in space and time, making its spatial variation difficult to assess with air quality monitoring instruments from ground-based networks or on-board of satellites (Hoek et al., 2015). Fixed monitoring stations are suitable for recording temporal variation of air pollution, including long-term trends, but not proper to capture the spatial variation of air pollution at local level (Li et al., 2019).

It has been demonstrated that gradients at urban scale can be identified by mobile monitoring (Deshmukh et al., 2020). High resolution mapping of air quality can be done based on long-term average of a significant number of repeated measurements (Upadhya et al., 2024). However, mobile monitoring to obtain reliable small-scale variations (for a street segment or a residential area) that are subsequently time-averaged to provide long-term concentrations is time-consuming, involving extended resources, due to the necessity to collect a large number of co-located data and in the same time to cover the whole relevant area (Yuan et al., 2024). Several models have been developed to overcome the weakness of limited availability of the observational data, either collected at fixed locations, or during mobile campaigns, or measured by instruments onboard of satellites. Data from ground-based mobile and fixed stations as well as satellites data have been used in Land-Use Regression (LUR) models

(e.g. (Apte et al., 2017; Anand and Monks, 2017; Messier et al., 2018; Shairsingh et al., 2019; Kerckhoffs et al., 2021, 2022a;
Xu et al., 2021b; Knibbs et al., 2018; Lee, 2019)) and dispersion models (e.g. Hamer et al. (2020); Ramacher et al. (2021); Snoun et al. (2023)). Both models have emerged as very promising and efficient tools for high-resolution mapping of the changes in the composition of the atmosphere, as well as for quantifying the air quality by long-term averaging at a high spatial resolution.

The LUR model is more widely used in air quality studies compared to dispersion modeling because: (a) it is a multivariate linear regression model built on significant covariates that can be further used to estimate pollutant concentration elsewhere, (b) linear regression is one of the most used fine-scale spatial interpolation methods because it is fast, easy to implement (Hoek et al., 2008; Jerrett et al., 2005), and does not require high computing power such as computational fluid dynamics based on large-Eddy simulation or Reynolds-averaged Navier–Stokes approaches (Lin et al., 2023, 2024), and (c) a LUR model does not require detailed information on atmospheric conditions and an emission inventory as input data. LUR model usually requires measurement data and land-use predictor variables (e.g. CORINE dataset) (Kerckhoffs et al., 2021, 2022b). Initially, LUR models were developed to estimate the concentration of air pollutants linked with traffic emissions, specifically $NO_2$ and $NO_x$ (Briggs et al., 1997; Stedman et al., 1997; Hoek et al., 2008; Eeftens et al., 2012; Lu et al., 2020; Zhang et al., 2021). Lately, LUR models have been successfully expanded to include other air pollutants, such as particulate matter (PMs) (Taheri Shahraiyni and Sodoudi, 2016; Karimi and Shokrinezhad, 2021; Zhao et al., 2021; Wallek et al., 2022), ozone ($O_3$) (De Marco et al., 2022; Wei et al., 2022), carbon monoxide (CO) (Bi et al., 2022), and sulfur dioxide ($SO_2$) (Wu et al., 2019; Mikeš et al., 2023). LUR models can now estimate a wide range of air pollutants, including black carbon (BC) (Xu et al., 2021a; Van den Bossche et al., 2015), volatile organic compounds (VOCs) (Zapata-Marin et al., 2022; Choi et al., 2022), and ultrafine particles (UFP) (Ge et al., 2022; Kerckhoffs et al., 2021; Lloyd et al., 2023; van Nunen et al., 2020; Jones et al., 2020; Saha et al., 2019). LUR models can be used both for specific sites, such as highways (Lee et al., 2013; Patton et al., 2014) or neighborhoods (Lim et al., 2019), as well as in detailed studies covering a wide range of land-use types across large city areas (Hatzopoulou et al., 2017; Van den Hove et al., 2020). Recent studies report on variations of pollutant levels across different times of the year, based on seasonal measurement campaigns (Xu et al., 2021a; Miri et al., 2019; Shi et al., 2020).

In general, LUR models tend to "smooth" concentration levels over a wide area, leading to under or overestimation of observed concentrations within each pixel. Therefore, one of the most feasible and robust approaches to map air quality at high resolution is to use the mixed-effects modeling framework that combines the advantages of measurement-only mapping and LUR modeling (Kerckhoffs et al., 2022a, b). Mixed-effects modeling is mostly used in scenarios where data is hierarchical or clustered (e.g. Fokkema et al. (2018); Seibold et al. (2019)). In air pollution research, mixed-effects models are powerful tools that can account for spatial or temporal clustering inherent in air quality data. They can accommodate factors like geographic regions or repeated measurements over time, providing a nuanced understanding of pollutant distribution. These models can be computationally complex, especially when dealing with large datasets or complicated random effects structures (Kerckhoffs et al., 2022a).

The mixed-effects model framework has been used in recent air quality studies for urban areas, like Amsterdam and Copenhagen (Kerckhoffs et al., 2022b), Oakland (US) (Kerckhoffs et al., 2024). Also, a similar mixed-effects approach has been

used to estimate $NO_2$ concentrations over Hong Kong (Anand and Monks, 2017). Up to now, no high-resolution mapping of air pollutants at high spatial resolution has been performed for urban areas in Romania, although many cities face serious atmospheric pollution episodes (Marin et al., 2019; Ilie et al., 2023).

In this paper we present the development and use of the mixed modeling framework for high-resolution mapping of $NO_2$, $PM_{10}$, $PM_{2.5}$ and UFP concentrations in Bucharest, the capital of Romania. Data from two mobile measurements campaigns, representative for warm and cold seasons, were combined with fine-scale land-use parameters to provide the spatio-temporal information necessary to predict seasonal surface concentrations. Results were validated against in situ measurements from Magurele Center for Atmosphere and Radiation Studies (MARS), and eight fixed observation stations operated by the National Air Quality Monitoring Network (NAQMN). A detailed description of the study area, measurements and data treatment are given in Section 2, the mixed-effects modeling framework tuning for Bucharest is given in Section 3, together with model performance evaluation and aggregated pollutants maps for warm and cold seasons in Bucharest.

## 2 Materials and methods

### 2.1 Study area

Bucharest is the most populated urban area, and the most important industrial and commercial center of Romania. According to the latest census, the population of Bucharest is approximately 2.1 million residents (INS, 2024), making it the sixth-largest city in the European Union by population. The city covers an area of about 240 square kilometers and has a dense urban structure. The land use of Bucharest is diverse (Figure 1), the central and northern parts of the city are predominantly residential areas, characterized by a mix of old and new housing developments. Most of the production sectors, such as machinery, textiles, chemicals, electronics, and business parks, all contributing significantly to the economic base of Bucharest, are located in the southern and western areas. (Ilie et al., 2023; Balaceanu et al., 2018). The surroundings of Bucharest are mostly agricultural areas and rural/pre-urban residential areas.

Located in the southeastern part of Romania, in the Romanian Plain, Bucharest has a humid continental climate, characterized by hot summers, cold winters and two short transitional seasons, spring and autumn. Due to atmospheric circulation patterns specific to the north mid-latitude zone, episodes of long-range transport of aerosols from desert regions (Sahara, Arabian Peninsula and Persia) and from wildfires can affect Bucharest's air quality. However, the major pollution sources are local, influenced by the topography, the different local-scale wind regimes and anthropocentric activities (Fenger, 1999; Grønskei, 1998; Marmureanu et al., 2017; Balaceanu et al., 2018; Marin et al., 2019; Ilie et al., 2023).

### 2.2 Observational data

Data was collected during two intensive mobile measurement campaigns carried out in Bucharest during May - July 2022 and January - February 2023. For each campaign, at least 15 measurement routes of approximately 100 km long and around 8 hours were carried out, under various meteorological conditions. In order to ensure consistent and quality data that highlight

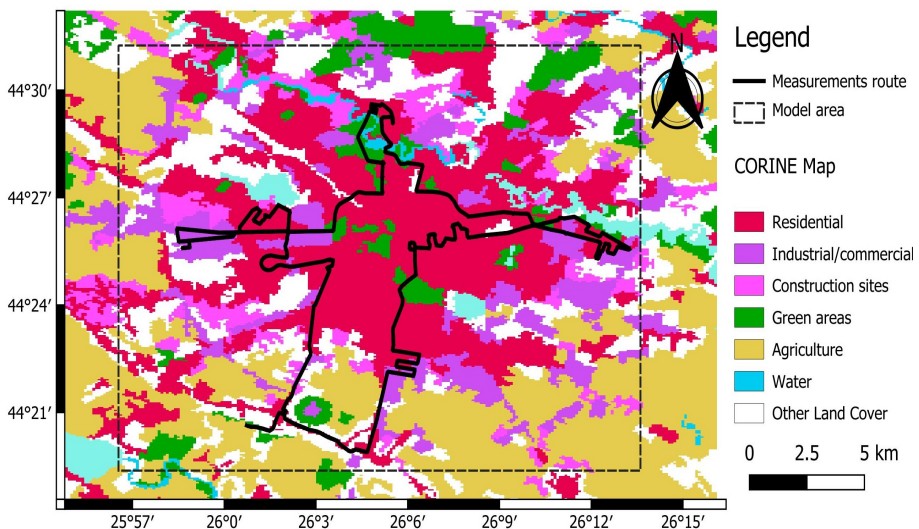

**Figure 1.** Landuse distribution in Bucharest, according to CORINE, 2018, black-overlapped mobile measurement route, dotted rectangular represent the modelled area

the variability of pollutants specific to warm or cold seasons, rainy and/or windy days were excluded from the measurement campaigns. Measurements were performed from Monday to Friday, from early morning to the afternoon, being therefore representative for daytime working times. The route comprised high-traffic streets, residential, industrial, and commercial districts, as well as suburban neighborhoods. The mobile measurement route is limited to the areas where the car has access, excluding some urban areas (e.g. parks, agricultural zones and water bodies). Portable equipment measuring UFP, particle matter fractions ($PM_1$, $PM_{2.5}$, $PM_{10}$), and $NO_2$ were employed in both campaigns, with a measurements rate of one second. An additional GPS system has been used to independently save the precise location. A Nafion dryer was also used during warm period to reach a humidity below 40% for UFP measurements.

The mobile data has been filtered before being ingested into the model. A moving average filter with a 3-data-point window was used to remove data points with values exceeding 1.5 times the window mean, above or below.

## 2.3 High-resolution mapping model

We used the land-use regression and mixed-effects modeling framework to develop high-resolution air quality maps for Bucharest. In a LUR model, the concentration of a pollutant is expressed as a linear combination of variables that approximates the influence of different emission sources and sinks. Usually LUR models are fixed-effect models because: (a) they use predictor variables that are temporally invariant (e.g. classes from the land cover inventories, statistical values of population density or traffic intensity, street networks) and, (b) they are applied to the average atmospheric state over the entire observation period. Therefore, LUR models based on fixed variables are not sensitive to unobserved heterogeneity arising from temporal variability in emissions or/and other environmental conditions.

The mixed-effects LUR model considered in this work is similar to the one developed in Kerckhoffs et al. (2022a). The daily mean concentration at a reference point $i$ on day $j$ ($Y_{ij}$) is assumed to be a linear function of the random effect ($A_{ij}$) and the predictor variables ($X_{ijk}$) computed at the same reference point:

$$Y_{ij} = \alpha_0 + \alpha_1 \cdot A_{ij} + \sum_{k \geqslant 0} \beta_k \cdot X_{ijk} + \in_{ij} \tag{1}$$

In Eq. (1), $\alpha_0$ is the random intercept, while $\alpha_1$ is the random slopes of $A_{ij}$, respectively. The $\beta_k$ are the regression coefficients of predictor variables $X_{ijk}$ at reference point $i$ and day $j$. The regression coefficients are the same for all measurements days. The error term of the model is represented by $\in_{ij}$. Mixed model results were averaged per point, similar to the average of the data-only approach. Mean $NO_2$, $PM_{10}$, $PM_{2.5}$ and UFP concentrations from each reference point were used as the dependent variables $Y_{ij}$ in Eq. (1).

The route taken during campaigns was divided into road segments with a length of approximately 250 m, equivalent to space traveled by a car with the average speed of 30 km/h in a time interval of 60 s. The reference points were considered at the midpoint of each road segment.

A temporal correction was applied on the data to synchronize the measurements. The correction factor is calculated for each point as the difference between the daily average values corresponding to the point and the whole-campaign average value corresponding to the same point. Values measured within a 250 m street segments were averaged for each day. Similar correction and averaging methods were reported in previous studies (e.g (Kerckhoffs et al., 2021, 2022a)).

The performance of the model has been evaluated in three steps. First, a subset containing 15% of data collected through mobile measurements (and not used to tune the model) was used for cross-validation. This percentage represents the optimal value for which the models developed in this study can recognize the relationship between the attributes of the input data and the output variable with $R^2$ score greater than 0.75. When selecting this percentage, providing as much quality data as possible (85%) was considered an important factor in the learning process to increase the performance of the model, as well as to avoid data leakage between the learning process and cross-validation. Second, an independent set of data collected at fixed sites was used for validation. In addition to the MARS site, eight monitoring stations operated by NAQMN in Bucharest were selected for validation, based on data availability, representing all types of environments: two urban-type stations located in the east and northeast of the city, one suburban station located 4 km to the south of the capital city, three industrial-type stations situated in the southern half of the city, and two traffic-type stations located in the center of the Bucharest. The evaluation involved direct comparison of statistical metrics (correlation, root mean square error, relative differences) between model outputs and pollutant direct measurements. The last step was the evaluation of the model performance to resolve different types of environment (traffic, urban, industrial). Details and results are provided in Section 3.1.

### 2.4 Tuning the mixed-effects Land-Use Regression model for Bucharest

Based on the specificity of the climate in the region of Bucharest, we decided to use the residential areas (predictor variable) as time-dependent variables with major differences between the warm season and the cold season (Ilie et al., 2023). Residential

areas are considered sources of pollution due to household activities, heating being responsible for the major difference between the cold and the warm seasons. However, the time dependence of this variable is not sufficient to describe the day-to-day variability because the residential heating is generally switched on in late autumn and off in late spring, with no real daily variability. To model $NO_2$, $PM_{10}$, $PM_{2.5}$ and UFP concentrations over the Bucharest area, an additional variable was needed in the LUR models to cover the fast time-dependencies (the so-called "random" effects). In this mixed-effects framework, the pollutant concentration can be expressed as a linear relationship between fixed variables and time-dependent variables, where the random effect is modeled by including a discrete "dummy" variable.

In our work, for each reference point, the "random effect" was modeled as the difference between the standard deviation calculated for the entire period of mobile measurements and the standard deviation calculated for each day of mobile measurements. Therefore, the magnitude and/or sign of the "random" effect were not the same over all reference points of measurements. The mixed-effects models were fitted to the observational dataset using the Python modules scipy, sklearn and statsmodels.

Other predictor variables used for the LUR models include vehicle traffic intensity (calculated separately for the warm and for the cold seasons) as well as aggregated values of spatial predictor variables calculated within circular buffers ranging from 25 m to 2 km in radius.

The mixed-effects LUR models (one model for each pollutant considered in this study) were adjusted and trained for Bucharest to obtain consistent datasets. For the training process, 85% subsets of mobile measurements were randomly selected. The remaining 15% of the mobile measurements were used to cross-validate the LUR models. By dividing the data set used in the learning process into 85% for training and 15% for testing, it was followed on the one hand, to increase the performance of the models developed in this study, and on the other hand, the aim was to reduce the overfitting effect of the models by obtaining the smallest possible difference between the $R^2$ score obtained during training and testing. The regression coefficients obtained as an output of the training were further used to generate high-resolution maps of seasonal concentrations of $NO_2$, $PM_{10}$, $PM_{2.5}$ and UFP with a resolution of 100 m, over an area of approximately 240 km$^2$ (the entire area of the city of Bucharest).

### 2.4.1 Spatial predictor variables

To define the optimal configurations of LUR architectures for Bucharest, for each 250 m street segment, spatial predictor variables were extracted using the following data sources: (i) CORINE for land cover (European Environment Agency, 2018), (ii) Open Street Maps (OSM) (OpenStreetMap, 2021) for road network and (iii) National Institute of Statistics (INS) for population density.

There is no recent source quantifying the traffic intensity on road segments in Bucharest, therefore the value for this predictor was estimated for each direction of the street segment $i$ using the following relationship:

$$Counts_i = \frac{N}{L} \cdot fc_i \cdot v_i \tag{2}$$

where N represents the total number of vehicles per day obtained from INS, L represents the total length of street segments (km), $v_i$ represents the speed on the street segment (km/h), $fc_i$ represents cost function for the street segment (h) retrieved from geofabrik.de database (Geofabrik, 2024).

In order to tune and train the mixed-effects models for the specifics of Bucharest, the spatial predictor variables were selected from a number of proxies that describe the possible sources and sinks of $NO_2$, $PM_{10}$, $PM_{2.5}$, $PM_1$ and UFP emissions. These variables are presented in Table 1. The column "Effect" represents the effect that the predictor variables have on the concentration of the pollutant in the atmosphere. Variables associated with emission sources have a positive effect, while variables associated with sinks, such as vegetation cover, have a negative effect. The circle radii most commonly used for buffering the variables describing the sources and sinks at a given location are given in column "Buffer sizes".

### 2.4.2 Predictor variable selection

For the selection of variables we first removed proxies with a percentage of null values greater than 90%. After applying this filter, proxies that describe the possible sources and sinks for LUR models tuned for Bucharest were associated with:

- traffic, with predictors "Counts" (number of vehicles per day) and road length variables in buffer of 50 to 1000 m

- land-use, with predictors: industry, green, residential lower density (individual residential), residential higher density (collective residential), construction and water in buffers of 100 to 2000 m

- population density in buffers of 100 to 2000 m

To determine the optimal combination of predictor variables, the supervised forward stepwise regression approach proposed within the European Study of Cohorts for Air Pollution Effects (ESCAPE) project (ESCAPE, 2024) was used. ESCAPE model starts from a constant value and after that the predictor variables are added based on the goodness of fit given by the adjusted cross-correlation ($R^2$) value. The direction of effect for all variables was kept as in Table 1. The variable with the highest adjusted $R^2$ was included in Eq. (1) as $X_k$. The process of building the model stops when the new variables do not contribute significantly (more than 1%) to the improvement of the adjusted $R^2$ value. The LUR model configurations were generated using all possible combinations of generated predictor values. From the total of the models obtained, only the LUR models for which the adjusted $R^2$ value was higher than 0.5 were selected.

In a second step we calculated the confidence level (p-value) and variance inflation factor (VIF) to identify that predictor's contribution to a collinearity problem. The statistically insignificant variables ($p > 0.05$) and predictor variables where VIF > 5 sequentially were not used in the model.

The predictor variables and the size of the buffers used in this work are shown in Table 1. The predictors passing the above conditions are shown in bold. The sizes of the buffers are those established within ESCAPE project for the development of LUR models. These buffer sizes are used to determine the spatial proximity of different features by defining a distance zone around the features.

Table 1: Description of spatial predictor variables

| Source data | Variable name | Description | Unit | Effect | Buffer sizes (m) |
|---|---|---|---|---|---|
| INS | **POPDENS_X** | Population density | $\#/m^2$ | + | 100, 150, 200, 250, 300, 500, **1000**, 2000 |
| CORINE 2018 | **LDRES_X** | Low-density residential | $m^2$ | + | 100, 150, 200, **250**, 300, 500, 1000, 2000 |
| | **HDRES_X** | High-density residential | $m^2$ | + | 100, 150, 200, 250, **300**, 500, 1000, 2000 |
| | AIRPORT_X | Airport area | $m^2$ | + | 100, 150, 200, 250, 300, 500, 1000, 2000 |
| | **INDUSTRY_X** | Industrial areas | $m^2$ | + | 100, 150, 200, **250**, 300, 500, 1000, 2000 |
| | AGRI_X | Agricultural areas | $m^2$ | +/- | 100, 150, 200, 250, 300, 500, 1000, 2000 |
| | **FOREST_X** | Forest areas | $m^2$ | - | 100, 150, 200, 250, 300, 500, 1000, **2000** |
| | GREEN_X | Urban green areas | $m^2$ | - | 100, 150, 200, 250, 300, 500, 1000, 2000 |
| | **CONSTR_X** | Construction sites | $m^2$ | + | 100, 150, 200, **250**, 300, 500, 1000, 2000 |
| | **WATER_X** | Water bodies | $m^2$ | +/- | 100, 150, 200, **250**, 300, 500, 1000, 2000 |
| Road network | **LENGTH_X** | Length of road segments | m | + | 25, 50, **75**, 100, 150, 200, 250, 300, 500, 1000, 2000 |
| | TRAFFIC_X | Total traffic load | (veh / day) * m | + | 25, 50, 75, 100, 150, 200, 250, 300, 500, 1000 |
| geofabrik.de; INS | **COUNTS** | Traffic intensity | veh / day | + | N/A |

# 3 Results and discussions

## 3.1 Evaluation of the model performances

For each individual pollutant, four configurations of LUR models were defined, out of those showing the adjusted $R^2$ greater than 0.5. Further, only the results and performances of the LUR models for which the highest adjusted $R^2$ value was obtained are discussed.

The mixed-effects model tuned for Bucharest city has been evaluated using mobile measurements and fixed-site measurements. The performance assessment involved not only the averages in some points, but also the overall agreement on specific
types of urban areas covering the entire pollutants concentration intervals. The performance of each model (one for each type of pollutant) was tested following three steps described in details in section 2.3. Moreover, in the first step, the 15% kept for testing covers all possible situations regarding the spatial distribution of the predictor variables used in the model. Second, the average concentrations calculated by the model at the location of the fixed observation sites have been validated by comparison with the seasonal-average concentrations measured at those sites. Also, model data clustered based on the type of the
environment (traffic, industrial, urban) has been compared to similarly clustered data collected at the fixed observation sites.

### 3.1.1   Cross-validation against mobile measurements

Cross-validation is performed for each model by comparing the model predicted dataset against the observed data collected by mobile measurements. Agreement between the two datasets is quantified by calculating the adjusted $R^2$ and root mean square error (RMSE). The RMSE of each model was calculated as the square root of the mean of the squared errors. We also
present the relative differences between modeled and measured data as a general indicator of the model accuracy. Results show very good correlations ($R^2 > 0.91$) for each model as a follow-up of the training process. The cross-validation shows higher correlations for the cold season ($R^2 = 0.81$ for $NO_2$ and $R^2 = 0.88$ for $PM_{10}$) than for the warm season ($R^2 = 0.59$ for $NO_2$ and $R^2 = 0.72$ for $PM_{10}$). The weaker performance of the models during the warm season can be explained by the large variability of $NO_2$ and $PM_{10}$ concentrations in the warm season compared to the cold season (Ilie et al., 2023), which is not completely
captured by our method. Moreover, this strong variability is also suggested by the RMSE values which for the $NO_2$ are higher in the warm season (2.94 ppb) than in the cold season (2.64 ppb). In contrast, the RMSE values for $PM_{10}$ are lower in the warm season (2.18 $\mu g/m^3$) than in the cold season (3.06 $\mu g/m^3$). A cross-validated scatter plots were also added in the supplement.

The relative differences between mobile measurements and model retrievals, computed as (Model-Observed)/Observed, show the ability of the models to estimate $PM_{10}$ and $NO_2$ concentrations (Fig. 2). It can be seen that the model for $NO_2$ tends
to overestimate the predicted concentrations in the warm season, especially in urban agglomeration areas (upper left panel), while the model for $PM_{10}$ tends to underestimate the predicted concentrations(lower left panel). However, such differences are acceptable considering the assumptions made, the uncertainty of the data used for training the models, as well as the general performances reported in the literature (Ma et al., 2024).

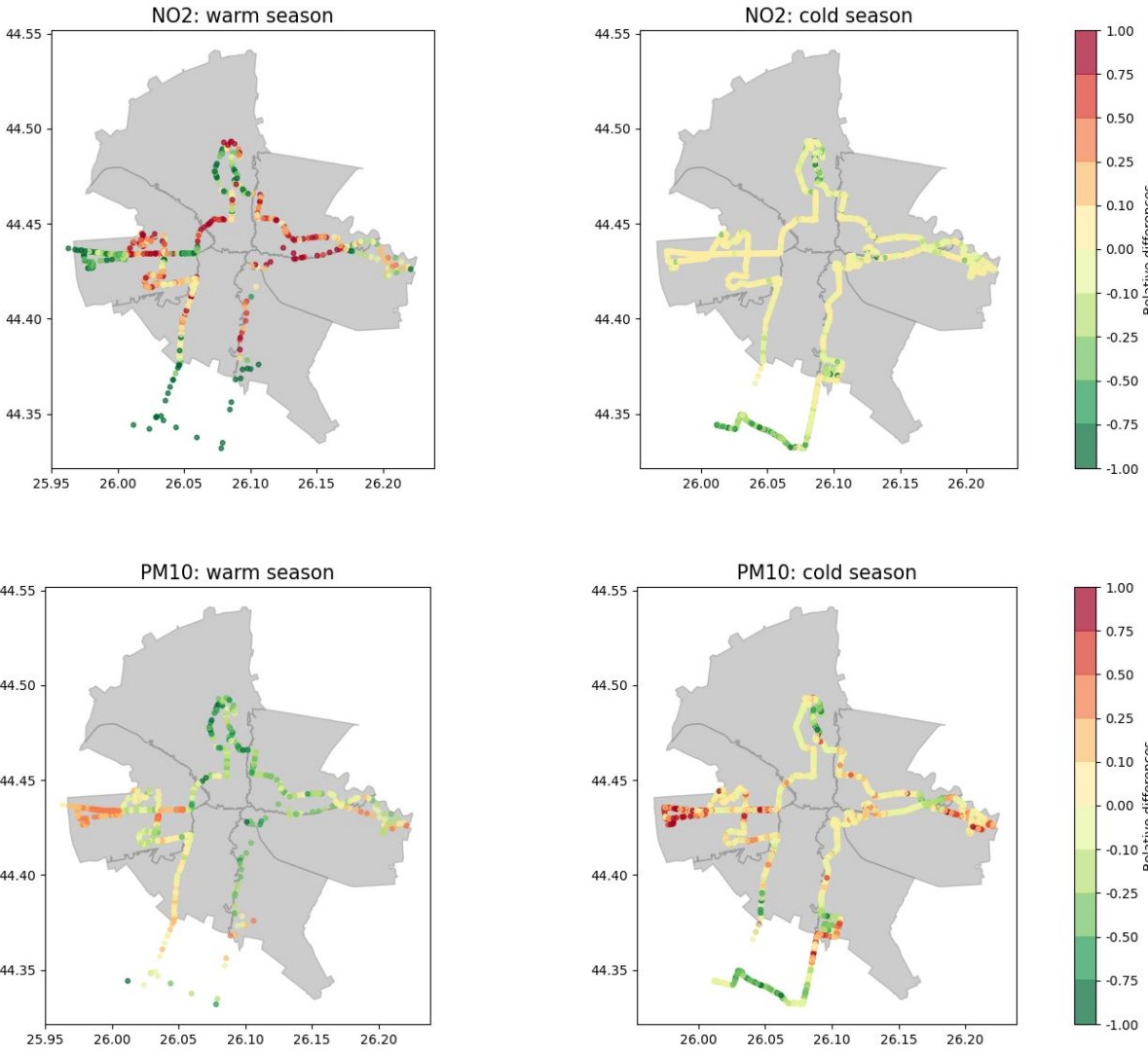

**Figure 2.** Relative difference between model predicted values and mobile measurements of NO$_2$ (upper panel) and PM$_{10}$ (lower panel) during warm (left panel) and cold (right panel) season

### 3.1.2 Validation against independent measurements at fixed observation sites

The model output was evaluated against the seasonal average of the hourly values of NO$_2$ and PM$_{10}$ measured at nine long-term operated in situ stations, out of which eight stations operated by NAQMN. These sites are considered representative for urban, industrial and suburban areas as pictured in Fig. 3. The model could not be evaluated in the case of UFP, due to unavailability of the data at all fixed stations. More information about what type of variables are measured by NAQMN are given in (Ilie et al., 2023). Detailed description of MARS site is given in (Pîrloagă et al., 2023), where continuous PM concentrations are

performed using optical particle counters (Mărmureanu et al., 2019) and gases analysers (Castell et al., 2018). Statistical metrics, like $R^2$, RMSE and relative differences were calculated similarly as for the cross-validation. Statistical parameters are summarized in Table 2.

In Fig. 3, the mean mass concentrations measured at the fixed observation sites are represented by the diameter of the circle, and the type of the environment is represented by the color of the circle. The missing data in Fig. 3 is due to the fact that no 275 measurements were available for these periods, due to various non-scientific reasons (technical problems with the measurement equipment, manpower problems, etc.) It can be seen that the variation of the concentrations across the city is relatively low, especially for particulate matter during the cold season. $PM_{10}$ concentrations range from 21 to 32 $\mu g/m^3$ in both seasons. The highest $PM_{10}$ concentrations are observed at the urban, sub-urban and the traffic stations, while the lowest $PM_{10}$ concentration is measured at the industrial stations. The western part of the city shows highest $PM_{10}$ values. $NO_2$ concentrations range from 280 8 to 20 ppb in the warm season and from 9 to 24 ppb in the cold season. The highest $NO_2$ concentrations correspond to the areas with intense traffic and industrial activities, while the lowest concentrations are observed in the sub-urban areas, which are less impacted by traffic.

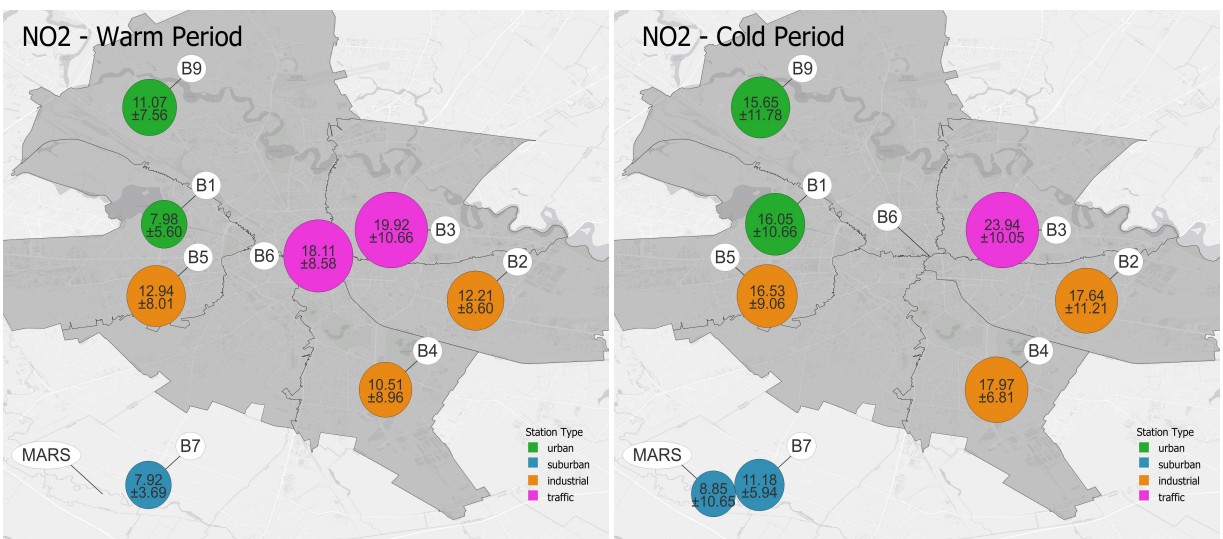

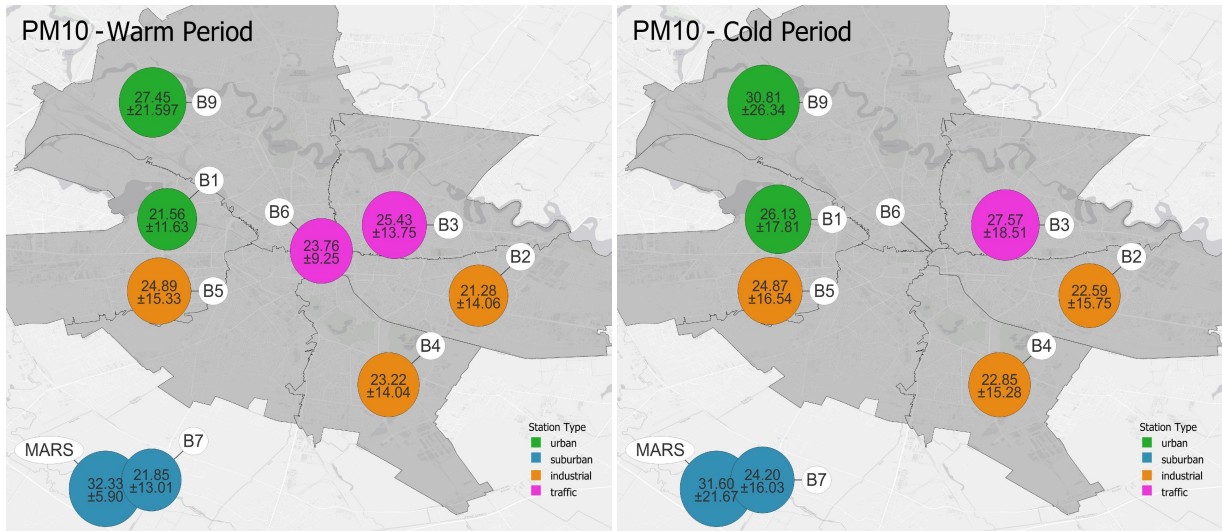

**Figure 3.** Average concentration of NO$_2$ (upper panel, units ppb) and PM$_{10}$ (lower panel, units $\mu g/m^3$) during warm (left panel) and cold (right panel) season at fixed sites: the diameter of the circle represents the mean mass concentrations measured at the site; the color of the circle represents the type of the environment at the site

.

The comparison between model predicted values and the observed values at the nine fixed sites is presented in Table 2. Overall, the model performed well, even if the NO$_2$ values tend to be slightly overestimated, and PM$_{10}$ tend to be slightly underestimated when compared with measured mean concentrations. Mean values of modeled NO$_2$ are within the range of the observed values, as indicated by the standard deviation. The differences can be explained by the local topography and the specifics of the land use. The road system in Bucharest is very dense, so the distance from a street to residential or industrial sectors is often very short, sometimes less than few meters, therefore the NO$_2$ 100 x 100 m grid resolution cannot always resolve the variations. This "smoothing" effect caused by insufficient spatial resolution of the model is pin pointed by the lower values of the standard deviations returned by the model by comparison with those returned by observations.

The trends and seasonal differences of both NO$_2$ and PM$_{10}$ are well resolved by the model, as shown by the comparison with observations. The R$^2$ correlation between observed mean concentrations and modeled mean concentration is above 0.65 for all pollutants for warm season, and 0.75 for cold season. These values can be attributed to the good performance of the model, in accordance with the values reported for other cities (Yuan et al., 2023). The lowest correlation value is noted for particulate matter during warm season. This result can be explained by the fact that, during the warm season, photo-chemical processes are intensified at the street level, and the model cannot capture the effects properly.

The RMSE is another important parameter for assessing the performance of the model, accounting for the level of absolute error. As in the case of R$^2$, we noticed a better model performance for the cold season. The difficulties of the model to capture

the small variations during the warm period for both $NO_2$ and $PM_{10}$ are depicted by higher RMSE values. High values of RMSE correspond to low $R^2$ values, demonstrating that the model cannot fully capture the variations of $NO_2$ or $PM_{10}$ concentrations.

**Table 2.** Comparison between model output and measured values at fixed sites for $NO_2$ and $PM_{10}$ in Bucharest (Romania) during warm period 2022 and cold period 2023, and statistical metrics

| Pollutant | Season | Observed mean concentration | Modelled mean concentration | $R^2$ | RMSE |
|---|---|---|---|---|---|
| $NO_2$ | warm | $12.58 \pm 7.71$ ppb | $16.38 \pm 2.47$ ppb | 0.66 | 4.97 ppb |
| | cold | $15.98 \pm 9.52$ ppb | $17.25 \pm 1.17$ ppb | 0.75 | 2.27 ppb |
| $PM_{10}$ | warm | $24.64 \pm 13.18 \ \mu g/m^3$ | $24.29 \pm 4.38 \ \mu g/m^3$ | 0.65 | $2.02 \ \mu g/m^3$ |
| | cold | $26.33 \pm 18.50 \ \mu g/m^3$ | $25.64 \pm 4.43 \ \mu g/m^3$ | 0.76 | $1.69 \ \mu g/m^3$ |

### 3.1.3 Evaluation of the model performance to resolve different types of environment

The model was tested for its robustness in capturing differences between various types of environment such as urban (including pre-urban), industrial, traffic. Long-term datasets collected at the nine fixed observation sites were used for this purpose. Figure 3 shows mean concentrations of $NO_2$ and $PM_{10}$ as retrieved by the model and measured at the stations, clustered by the type of environment and separated for the warm and cold seasons. The relative differences between values modeled and measured in different environments are also highlighted.

$NO_2$ is the most variable species, with high differences between seasons and between environment types. Lower $NO_2$ concentrations are depicted for the urban group, whereas the highest $NO_2$ concentration is associated to traffic, as anticipated. The traffic group has the lowest variability among seasons and is also better captured by the model. The model shows increased $NO_2$ concentrations during the warm season for urban and industrial categories, while in the case of traffic areas the model underestimate a bit the measurements averages. The overall relative differences between cold and warm seasons for both model and observational data inside each defined environment is less than 35%. Higher relative differences between modeled and measured data are observed for warm season for all areas, with lowest difference in the case of traffic areas, around 10%. Overall lower relative differences between modeled and measured data are observed during cold season in comparison with warm period, with $NO_2$ average concentration in industrial and traffic areas underestimated by the model, highlighted by relative differences up to -20%.

$PM_{10}$ concentrations show an overall lower seasonal variability, and also lower differences between model and observed data, with relative differences less than 20%. Particulate matter concentrations for urban, industrial, and traffic areas varied slightly among seasons. Modeled $PM_{10}$ concentrations shows higher values for industrial environment in comparison with observational data for both seasons. Moreover, industrial areas present the lowest $PM_{10}$ concentration during cold season, as shown by both modeled and measured data. Urban and traffic environment $PM_{10}$ concentrations are slightly underestimated by the model. The overall relative differences of $PM_{10}$ concentration between cold and warm seasons for both model and

observational data inside each defined environment is less than 15%. Small relative differences between modeled and measured data are observed for $PM_{10}$ concentrations, during both seasons. $PM_{10}$ average concentration in urban and traffic areas are slightly underestimated by the model, highlighted by relative differences up to -10%, which can be also influenced by the low number of available fixed stations in each environment.

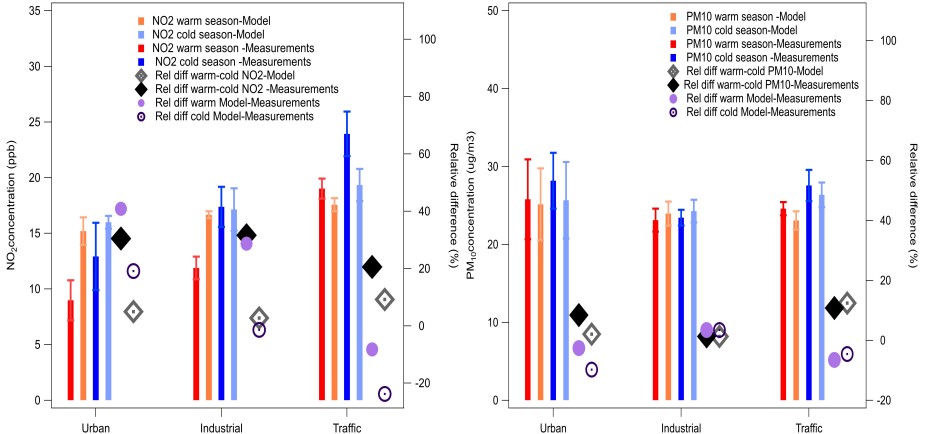

**Figure 4.** Model (light color) versus measurements (dark color) mean concentrations of $NO_2$ (left panel) and $PM_{10}$ (right panel) during warm (red) and cold (blue) seasons, along with the relative difference between cold and warm season from model (grey diamond mark) or measurements (black diamond mark); relative difference between model and measurements for warm (purple circle mark) and cold (dark purple circle mark) seasons

## 3.2  Mapping atmospheric pollution in Bucharest

The validated model was further used to produce $NO_2$, $PM_{10}$ and UFP concentrations maps for Bucharest, representative for the warm and cold seasons (Fig. 5). The results are valid for daytime and working days of the week. It should also be taken into account that, in the absence of spatio-temporal emission inventories with a high spatial resolution and traffic data, modeled data were used. In analyzing these results, it must be noted that the near-surface concentrations of atmospheric pollutants are influenced not only by the emissions, but also by the height of the planetary boundary layer, transport from other regions, dry and wet deposition and chemical processes, all in relation to relatively fast changing meteorological conditions (e.g. air temperature, wind field). Model results show that, overall and regardless the season, $NO_2$, $PM_{10}$ and UFP concentrations are higher on the main road sections, with higher values on Bucharest's western area.

$NO_2$ concentration maps show that this pollutant is highly related to traffic, the road network of Bucharest being clearly visible both during the warm and during the cold season. The main roadways, especially the Bucharest ring road, are depicted as the primary $NO_2$ source. Also, featured are the city central routes, where traffic remains heavy throughout the day and seasons. The highest $NO_2$ concentrations is noted around busy highways due to the presence of a large number of NO-emitting automobiles. Sinks related to the green areas and water bodies regions are identified in dark green colors. Overall, $NO_2$ con-

centration is higher during the warm period, when concentrations are higher on key roadways (35.79 ± 8.38 ppb) and other sources in the city add up. The conversion of NO to $NO_2$ in the presence of sunlight and ozone is significant. During cold months, the $NO_2$ concentration is lower in absolute values across the city, however the main roads are still depicted as major sources, followed by several industrial areas such as power plants serving the centralized heating of the city. The distribution of the $NO_2$ concentration on all street segments is almost uniform during cold season, with a slight increase on the main street segments, including the Bucharest ring road (20.67 ± 3.44 ppb). At the level of the city of Bucharest, the average value of the $NO_2$ concentration as estimated by the model for the cold season is 16.66 ± 4.04 ppb, while for the warm season it is 18.75 ± 1.98 ppb.

The spatial variation of PMs in the city area is substantial, with an abundance of small particles and a high mass concentration of larger particles in densely populated residential areas. Significant concentrations of particles have been identified, mostly in industrial areas and anthropogenic agglomerations, but also along certain major transportation routes. During the cold periods, the PMs (all sizes) have larger loading and lower gradients, as reported also for other cities (Ndiaye et al., 2024). This is related to increased emissions from residential heating. A gradient of $PM_{10}$ concentrations is evidenced within the city, with higher loading in the western and southern areas. Average $PM_{10}$ concentration in Bucharest during cold season is 1.2 times higher than in the warm season. During the warm periods, the $PM_{10}$ clusters are localized around source areas, while during cold periods, the sources distribution are more homogeneous.

The average UFP number concentration throughout the mobile route exhibits an important spatial gradient, particularly during the warm season, with variations up to a factor of two in the mean, highlighting extensive human exposure to ultra-fine particles. The UFP number concentrations are elevated on the main roads, as well as on some areas related to industrial activities in southern, northern and western city regions, mostly during the warm season. A more uniform distribution of UFP mean number concentration is observed during the cold season, when the house heating emissions add up to the traffic. Traffic sources have less impact during the cold season, as chemical processes diminish due to limited sunlight. House heating sources, more evenly distributed at spatial scale than the roads, generate more homogeneous distribution but also larger absolute values of UFP. Average seasonal concentration on Bucharest city during the cold season (29132 ± 4362 particles/cm$^3$) is 1.4 times higher than in the case of warm season (21469 ± 3528 particles/cm$^3$).

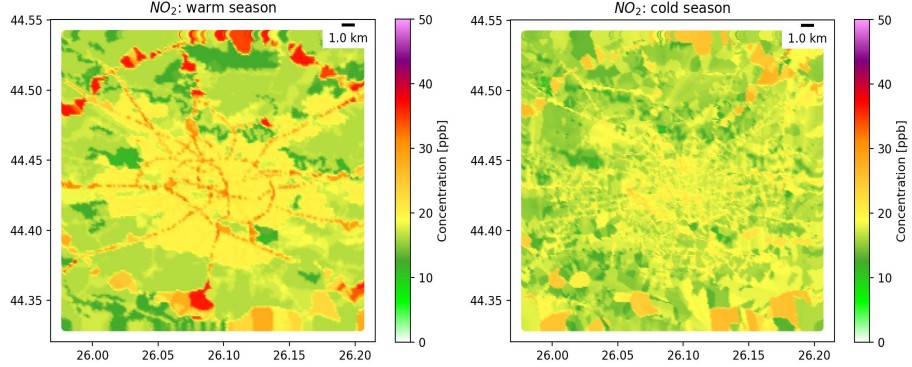

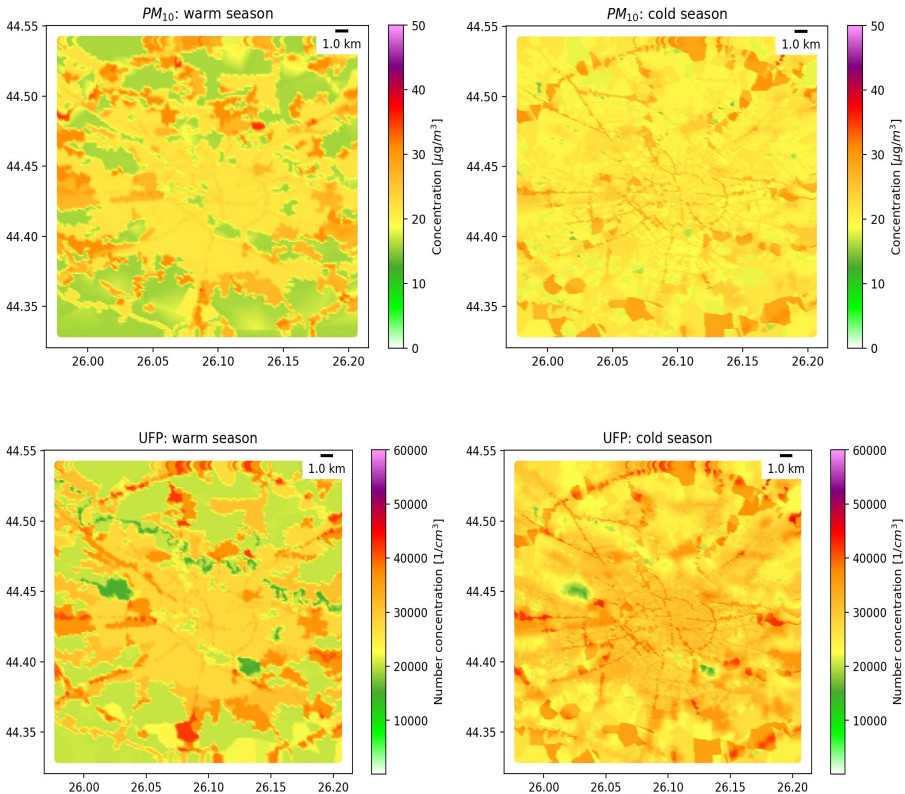

**Figure 5.** Near-surface concentration maps for Bucharest, as resulted from the model for $NO_2$, $PM_{10}$ and UFP during warm (left panel) and cold (right panel) seasons

Since measurements of $PM_{2.5}$ were only available at a few fixed stations, we included the modeled ratio of $PM_{2.5}$ to $PM_{10}$ as the result to show what we expect for the fraction of fine particles from the model. The model shows the fine particle fraction ($PM_{2.5}$ / $PM_{10}$) to be larger during the cold periods, compared to warm periods, with fine particles accounting for up to 95% of the $PM_{10}$ concentration (Fig. 6). This is explained by the fact that household activities generate predominantly small particles and higher percentages are seen in the peri-urban regions (outside of Bucharest) where the house heating are contributing more (lighter color of purple, Fig. 6 right panel) to $PM_{2.5}$ concentrations. During warm periods, fine particle fraction is approximately 50% within the city and less than 40% in the villages close to Bucharest where the agricultural activities increase the $PM_{10}$ fraction (Fig. 6 left panel). The main rivers and lakes within Bucharest's perimeter are clearly sinks for small particles, producing lower fine mode fractions in both seasons.

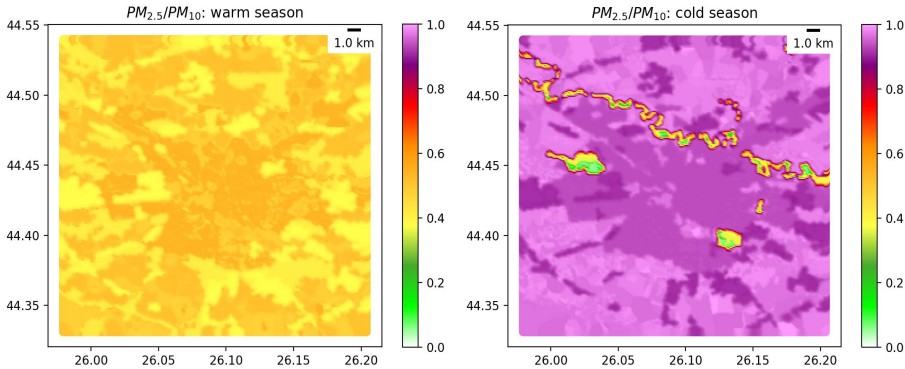

**Figure 6.** Model maps of $PM_{2.5}/PM_{10}$ ratio during warm and cold seasons

## 4 Conclusions

The regression-based methods fed by mobile data can predict $NO_2$, $PM_{10}$ and UFP concentrations for regions which are not properly covered by observations. In order to do this, the right combination of data sampling frequency, duration and route, and the correct number and type of predictor factors (corresponding to the surrounding environment) must be considered. Mobile monitoring together with modeling tools can therefore compensate for spatial and temporal data gaps which are collected by the monitoring stations, and can assist individuals and policymakers in identifying regions and causes of poor air quality. In this study we demonstrate the effectiveness of combining mobile measurements with mixed-effects LUR models to derive seasonal maps of near-surface $PM_{10}$, $NO_2$ and UFP. The study shows first-time validated results for Bucharest, the capital city of Romania, which is characterized by a large, densely populated surface with a very dense and heavily used street network.

Despite the limited number of fixed stations available for this work (8 + MARS), the tuned mixed-effects LUR model proved to be robust and accurate in producing high-resolution mapping of $NO_2$, $PM_{10}$ and UFP for the warm and the cold seasons. Overall, good model performance was observed for both seasons and all concentrations, similar to other studies. The slightly higher mean squared error values coupled with smaller cross-validation $R^2$ values obtained for the warm season suggest that the mobile campaign data collected for this study did not capture all the important $NO_2$ and $PM_{10}$ concentration variations. Even if the route selected for the two mobile measurement campaigns included all urban structures, the limitation of car access remains a source of error, which can lead to an underestimation of the concentration of pollutants. The performance of this model can be greatly improved by the involvement of citizens (pedestrian or by bicycle) to collect data from areas where cars are not allowed. Data sets systematically collected by citizens during daily (repeated) activities or walks could provide improved estimates of spatial variability for these areas (Snyder et al., 2013; Hankey and Marshall, 2015; Van den Bossche et al., 2016, 2018).The citizen involvement increases the pollutants data collection on areas restricted for cars or bicycles and enable the possibility to study the sinks on green or water areas, but are based only on low cost sensors. Further improvements could also be made by the inclusion of spatio-temporal emission data with a high spatial resolution, such as traffic volumes or emission inventories.

The results provided by the model show that high concentrations of particulate matter during the cold season are representative for Bucharest city, due to the added effect of house heating (either dispersed in residential areas, or localized at the city's power plants). Fine particles dominate during the cold season, although they remain at high levels during the warm season as well. $NO_2$ is less challenging but still an important factor, especially during the warm season and along the main roads. The seasonal high-resolution air quality maps for Bucharest based on mixed-effects modeling pinpointed pollutant variability mostly during the warm season and higher concentrations and fine particles ratio during the cold season. Water and vegetation areas are evidenced as effective sinks for $NO_2$ and fine particles, while traffic and residential heating are evidenced as effective sources in Bucharest. Based on these findings, a more extended certified AQ station networks would be beneficial for human health related pollutants monitoring, as well as inclusion of fine particles measurements among these

The approach presented in this paper can be adjusted for high-resolution mapping of $NO_2$, PMs and UFP in other cities as well,using the series of predictor variable identified in this study as necessary. This is feasible as long as the urban structures are well-characterized and there is a fairly dense and diverse network of in situ monitoring stations, whose observational data can be used for model calibration and validation.

*Code availability.* Codes developed for this study are available from the main author (Camelia Talianu, camelia@inoe.ro) upon request.

*Data availability.* Data used in this study are available from the main author (Camelia Talianu, camelia@inoe.ro) upon request.

*Author contributions.* **Camelia Talianu**: Conceptualization, Methodology, Formal analysis, Software, Writing – original draft, Writing – review & editing. **Jeni Vasilescu**: Conceptualization, Formal Analysis, Methodology, Resources, Supervision, Writing – original draft, Writing – review & editing. **Doina Nicolae**: Conceptualization, Funding acquisition, Project administration, Resources, Supervision, Writing – original draft, Writing – review & editing. **Alexandru Ilie**: Data curation; Investigation; Visualization; Writing – original draft. **Andrei Dandocsi**: Formal analysis, Investigation; Writing – original draft. **Anca Nemuc**: Writing – review & editing. **Livio Belegante**: Data curation; Investigation; Formal analysis.

*Competing interests.* The authors declare that they have no competing interests

*Acknowledgements.* We thank INCAS and INOESY for making available the mobile instruments. Also, we thank BOKU-Met, Vienna, Austria for facilitating access to the supercomputer from the IT infrastructure used for the LUR models. This publication has been prepared using European Union's Copernicus Land Monitoring Service. We gratefully acknowledge the National Air Quality Monitoring Network,

part of the Ministry of Environment and the Romanian Institute of Statistics for providing invaluable data essential to this research. We appreciate the anonymous reviewers and editors for their time and effort to improve the paper.

**Funding**

This work was carried out through RI-URBANS project (Research Infrastructures Services Reinforcing Air Quality Monitoring Capacities in European Urban & Industrial Areas, European Union's Horizon 2020 research and innovation program under grant agreement, Grant Agreement number 101036245), ATMO-ACCESS project (European Commission under the Horizon 2020 – Research and Innovation Framework Programme, H2020-INFRAIA-2020-1, Grant Agreement number: 10100800) and
the Core Program within the National Research Development and Innovation Plan 2022–2027, carried out with the support of MCID, project no. PN 23 05.

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
