# Peer review of "High-resolution air quality maps for Bucharest using Mixed-Effects Modeling Framework"

_EGUsphere, 2024_

## Author Comment (AC1)

Dear referee,

Thank you very much for the valuable comments to our paper and suggestions for article improvements. The manuscript has been modified accordingly.

Below are the answers and modification on the manuscript. In the following: "RefC" is the comment from Referee, "AuthR" is the author's response and "AuthCM" represents the author's changes to the manuscript. Page and line number refer to the page and line number in the version submitted for discussion.

**Specific comments**

**Abstract**

**Comment 1.**

RefC: "Lines 1-3: Please consider revising the first sentence of your abstract to improve clarity. I would suggest breaking the long sentence into shorter ones. Please consider using a single, consistent term (e.g., "high-resolution mapping" or "fine-scale mapping") throughout the manuscript to ensure uniform terminology and avoid potential confusion. Otherwise, the bulk of the abstract seems to be well constructed."

AuthR: The recommended corrections were done in the text. Also, "fine-scale mapping" term changed throughout the paper with "high-resolution mapping".

AuthCM: Page 1, lines 1-3: First sentence of the abstract revised: "High-resolution mapping of pollutants based on mobile observations facilitates deep understanding of air pollutant distribution within a city. This approach fosters science-based decisions to improve air quality, by adding up to the existing but not optimally distributed permanent monitoring stations."

**Introduction**

**Comment 1.**

RefC: "Lines 19 – 21. Please consider expanding this paragraph. Short term exposure to very high concentrations can also be a significant risk factor to human health in addition to prolonged exposure at lower concentrations."

AuthR: Modified according to the reviewer's note

AuthCM: Page 2, line 19: added text "Short term exposure to very high concentrations can also be a significant risk factor to human health in addition to prolonged exposure at lower concentrations."

**Comment 2.**

RefC: "Lines 23 – 25. "Despite its critical impact," I believe this paragraph may benefit from a relevant citation."

AuthR: Modified according to the reviewer's note

AuthCM: Page 2, line 23: added reference (West et al, 2016)

**Comment 3.**

RefC: "Also line 35, lines 35-36 and 36-39 should be properly referenced."

AuthR: modified according to the reviewer's note

AuthCM: Page 2, lines 35-39: relevant citations added for line 35 "(Deshmukh et al., 2020)", for lines 35-36 "(Upadhya et al., 2024)" and for lines 36-39 "(Yuan et al., 2024)

**Comment 4.**

RefC: "Lines 39 – 46. "Several models have been developed..." Apart from LUR and dispersion models, some additional well-established models and methods used for in urban air quality studies can also be mentioned in this section. The LUR is sufficiently detailed in the following paragraphs however some other methods may be worth mentioning."

AuthR: There are of course many models and methods used for air quality. In our article, we mentioned the models close to our approach.

AuthCM: none

**2 Materials and methods**

**2.1 Study area**

**Comment 1.**

RefC: Lines 90 – 95: "The land use of Bucharest is diverse" ....
Please consider adding an additional figure showcasing the diverse land use types present in the Bucharest metropolitan area. Additionally, one may plot the routes covered in the measurement campaign to showcase the lengths covered in each of the major land use areas. This would allow the readers to better assess the spatial coverage of the study in relation to Bucharest's heterogeneous urban structure.

AuthR: Added Figure 1, according to the reviewer's note

AuthCM: A map with diverse land use types present in the Bucharest metropolitan area and the routes from the measurement campaign has been added as Figure 1.

**2.2 Observational data**

**Comment 1.**

RefC: The authors describe conducting 15 measurement routes per campaign, each approximately 100 km in length. While the spatial coverage across different urban typologies is noted, it would be valuable to elaborate on the rationale behind this specific campaign structure. What were the key factors determining the number and configuration of routes? Statistical requirements for the modelling? Logistical constraints (traffic patterns, vehicle range, or time limitations)?

AuthR: The campaign timeline and structure were influenced by several key considerations:

Campaign Timeline and Time Constraints: Each route required several hours to complete due to the need for detailed measurements along the designated track. This constraint limited the number of routes to one per day. To ensure consistency and comparability, each route was conducted during the same time period on different days, allowing relevant statistical conclusions to be drawn without introducing temporal biases.

Seasonal and Meteorological Limitations: The number of measurement routes was also limited by weather conditions, as rainy days had to be excluded to avoid disruptions in data collection. Furthermore, only measurements performed under similar seasonal conditions were considered, ensuring that the dataset accurately captured pollutant variability specific to the warm or cold seasons.

Configuration of Routes: The configuration of the routes was designed to comprehensively cover the entire city while considering key factors: a) Pollutant Sources and Dispersion: Routes were planned to capture

areas with major emission sources and account for pollutant dispersion directions. b) Urban Typologies: The campaign ensured coverage of all relevant urban area types (e.g., residential, industrial, traffic-heavy zones) to meet the requirements of the dispersion model and provide representative data for the mixed-effects LUR model. This approach ensured that the campaign provided a robust dataset for high-resolution pollutant mapping while accounting for logistical, meteorological, and statistical considerations.

AuthCM: Page 4, line 105: added in text "In order to ensure consistent and quality data that highlight the variability of pollutants specific to warm or cold seasons, rainy and/or windy days were excluded from the measurement campaigns."

**Comment 2.**

RefC: Lines 111-113. Please elaborate on the decision to use the pass-band filter with a window of 3 data points. What advantages are expected for choosing this smaller window? Why not use a larger window of 5 to 10?

AuthR: The 3-points window was chosen due to the spatial relevance, the pollutants concentrations are similar in a given area, which is rapidly passed by the moving car. To be more clear to the reader the text has been adjusted.

AuthCM: Page 4, line 111: "3 data points was used to remove data-points with values higher and lower than 1.5 times the window mean." was changed to "3 data points window was used to invalidate the data-points with values higher and lower than 1.5 times the window mean."

**2.3 Fine-mapping model**

**Comment 1.**

RefC: Lines 131 – 137. The authors explicitly address a temporal correction, however it is not clear to me if this is enough to cover for traffic congestions. I would expect that longer instances of traffic congestion and/or lower than intended traveling speeds would result in skewed measurements for any given 250m segments. How do the authors address this issue? Are there additional temporal corrections considered?

AuthR: Each route in the measurement campaigns was carried out in the same time period on different days, allowing consistent measurements for each street segment without the introduction of temporal bias due to "traffic congestion". For this reason, we considered that an additional temporal correction is not necessary to cover "traffic congestion".

AuthCM: none

**Comment 2.**

RefC: Lines 139-140: "First, a subset of data collected. . ."
How large was the subset of the cross-validation data? 20-30%? And what was the reasoning behind this percentage? Please elaborate on this topic.

AuthR: In this study a percentage of 15% of the data was selected for cross-validation. This percentage represents the optimal value for which the models developed in this study can recognize the relationship between the attributes of the input data and the output variable with $R^2$ score greater than 0.75. When selecting this percentage, providing as much quality data as possible (85%) was considered an important factor in the learning process to increase the performance of the model, as well as to avoid data leakage between the learning process and cross-validation.

AuthCM: Page 6, line 138: "First, a subset of data collected . . . " was changed to "First, a subset containing 15% of the collected data . . . ."
Page 6, line 139: added text "This percentage represents the optimal value for which the models developed

in this study can recognize the relationship between the attributes of the input data and the output variable with $R^2$ score greater than 0.75. When selecting this percentage, providing as much quality data as possible (85%) was considered an important factor in the learning process to increase the performance of the model, as well as to avoid data leakage between the learning process and cross-validation."

**3 Results and discussions**

**3.1 Tuning the mixed-effects Land-Use Regression model for Bucharest**

**Comment 1.**

RefC: Lines 163-165. How was the traffic intensity affected on rainy days? Was there any substantial rain fall during the measurement campaigns? One would expect an increase in traffic intensity leading to an increase in the random effect. Can the authors elaborate on this topic? Was the model performance impacted by higher traffic intensity on rainy days (if such conditions were present)?

AuthR: The description was added to the text, explanations on Comment 7 response.

AuthCM: none

**Comment 2.**

RefC: Lines 167-168. I see the 85/15% split in training/validation is presented here. How did the authors tackle any overfitting issues that may arise from this split? Where any other splits considered and if so, what led to this split choice? Why is this optimal for your study? Please elaborate.

AuthR: The problem of overfitting was addressed by analyzing the $R^2$ score obtained during model training and testing. In this analysis, the objective was to obtain a score as high as possible after the training process by allocating a large number of consistent and quality data, as well as obtaining the smallest possible differences between the $R^2$ score obtained during training and testing. Splitting the data set used in the learning process into 85% for training and 15% for testing was considered optimal because in this scheme the smallest differences between the $R^2$ training score and the $R^2$ testing score ( below 0.05) and a $R^2$ score greater than 0.5 after cross-validation applied for all models developed in this study.

AuthCM: Page 6, line 168: added text "By dividing the data set used in the learning process into 85% for training and 15% for testing, it was followed on the one hand, to increase the performance of the models developed in this study, and on the other hand, the aim was to reduce the overfitting effect of the models by obtaining the smallest possible difference between the $R^2$ score obtained during training and testing".

**3.1.1 Spatial predictor variables**

**Comment 1.**

RefC: Comment: I am aware that the 2018 CORINE land cover data is currently the most recent iteration, however I suspect that the metropolitan area in Bucharest may have suffered some changes in the last 6 years since this dataset was made public. To this extent the additional map requested for section 2.1 may be useful for representing the land cover types and the campaign routs. One would expect some inconsistencies in the land cover data if any routs where traversing outside the residential areas.

AuthR: Modified according to the reviewer's note, see the answer to Comment no. 6.

AuthCM: none

**Comment 2.**

RefC: Line 176: "There is no recent source quantifying the traffic intensity on road segments in Bucharest..."
What does "recent" mean for the authors? To my knowledge, some municipalities and/or local police agencies should have some up to date traffic data, including vehicle types and population data. Is this data not publicly available in Bucharest?

AuthR: Unfortunately, we did not find official traffic data for Bucharest released for past period.

AuthCM: none

**Comment 3.**

RefC: Line 184: "... in Table 1. The column "direction of effect" ....
The specified column is labelled simply as "effect". For clarity please choose to update either the table column or the text at line 184.

AuthR: The recommended corrections were done.

AuthCM: Page 7, line 184: changed to "... Effect."

**Comment 4.**

RefC: Also, please update the source data column for the traffic intensity variable in table 1.

AuthR: The recommended corrections were done.

AuthCM: Table 1: The source data column for the traffic intensity variable was updated with "geofabrik.de; INS"

**3.1.2 Predictor variable selection**

**Comment 1.**

RefC: Line 198: "The direction of effect for all variables was kept as in Table 1"
If this is indeed the case, what is the reasoning behind the +/- effect for "Agricultural areas" and "Water bodies"? Later we find out that water bodies were obvious sinks.

AuthR: Agriculture and water bodies are both a source and sink of greenhouse gases (GHGs) and particulate matter (PM) due to anthropogenic as well as natural drivers (IPCC - Climate Change and Land, 2019). As we mentioned in the text, when defining the predictor variables, we took into account both the effects of agricultural areas and water bodies. After applying the selection criteria of the predictor variables, it resulted that for this study the water bodies only have a sink effect.

AuthCM: none

**3.2 Evaluation of the model performances**

**Comment 1. Part A**

RefC: Lines 215-216: Please give some additional on the criteria for selecting the 15% cross-validation data set in order to assess if there may be any data leakage present.

AuthR: This percentage represents the optimal value for cross-validation process for which the models developed in this study can recognize the relationship between the attributes of the input data and the output variable with $R^2$ score greater than 0.5. To avoid a possible data leakage between the learning and validation process, the data selected for cross-validation were not used in the learning process.

AuthCM: Page 9, line 216: added text "The percentage of 15% kept for testing represents the optimal value for which the models developed in this study can recognize the relationship between the attributes of the input data and the output variable with $R^2$ score greater than 0.5. Moreover, it cover all possible situations regarding the spatial distribution of the predictor variables used in the model."

**Comment 1. Part B**

RefC: Was the data selected from all routes and segments?

AuthR: yes, the data was selected from all routes and segments.

AuthCM: none

**Comment 1. Part C**

RefC: Was there any consideration given to any potential data leakage from the training to the cross-validation data?

AuthR: yes, to avoid data leakage between training and validation, the data selected for cross-validation were not used in the training process (see line 216: "...the 15% randomly selected mobile datasets not used for training ...)

AuthCM: none

**Comment 2**

RefC: Lines 229-231: I am not sure how to interpret the RMSE values without any mean values. This is just a personal preference but for me, a simple scatter plot would have been more helpful.

AuthR: Added scatter plots as supplement.

AuthCM: Page 10, line 231: added text "A cross-validated scatter plots were also added in the supplement."

**Comment 3.**

RefC: Figure 1 could be improved by adding an additional layer. An open street map or an RGB satellite layer should improve the interpretation of these relative differences. I see figure 2 has one such layer so maybe keep the same theme.

AuthR: The recommended corrections were done.

AuthCM: Figure 1 (now Figure 2) has been modified as recommended.

**3.2.2 Validation against independent measurements at fixed observation sites**

**Comment 1.**

RefC: Line 240 – 242: The authors mention that UFP data is not available in the NAQMN. Some additional information about what type of variables are/can be measured by NAQMN may be helpful for the readers.

AuthR: Done, see changes to manuscript.

AuthCM: Page 11, line 242: added text "More information about what type of variables are measured by NAQMN is given in (Ilie et al, 2023)"

**Comment 2.**

RefC: Please update figure 2 according to the type of variable being displayed. Both the upper panel and the lower panel indicate NO2, however the authors mention PM 10 concentrations in the Figure description.

AuthR: The recommended corrections were done.

AuthCM: Figure 2 (now Figure 3) has been modified as recommended.

**Comment 3.**

RefC: Line 244: Please explain the missing data in figure 2. For example, why are NO2 concentrations missing from B6 – cold period, or from MARS in the warm period. The same for PM10.

AuthR: No measurements were available for these periods, due to various non-scientific reasons (technical problems with the measurement equipment, manpower problems, etc.)

AuthCM: Page 11, line 245: added text "The missing data in Fig. 3 is due to the fact that no measurements were available for these periods, due to various non-scientific reasons (technical problems with the measurement equipment, manpower problems, etc.)"

**Comment 4.**

RefC: Table 2. A general comment also applicable to the remainder of the manuscript.
If both PM2.5 and PM10 were modelled, why are PM2.5 missing form the results section? How did the model perform against the independent measurements of the NAQMN? If the data is available why not at least update table 2 with PM2.5.

AuthR: $PM_{2.5}$ measurements are available only at few fixed stations. We included the modeled ratio between $PM_{2.5}$ and $PM_{10}$ to show what we expect for the fine particle fraction from the model. In the revised manuscript Fig.5 becomes Fig.6. For this reason, in the added text in the manuscript, Fig. 6 is mentioned.

AuthCM: Page 16, line 333: added text "Since measurements of $PM_{2.5}$ were only available at a few fixed stations, we included the modeled ratio of $PM_{2.5}$ to $PM_{10}$ as the result to show what we expect for the fraction of fine particles from the model."
Page 16, lines 333-337: changed to "The model shows the fine particle fraction ($PM_{2.5}$ / $PM_{10}$) to be larger during the cold periods, compared to warm periods, with fine particles accounting for up to 95% of the $PM_{10}$ concentration (Fig. 6). This is explained by the fact that household activities generate predominantly small particles and higher percentages are seen in the peri-urban regions (outside of Bucharest) where the house heating are contributing more (lighter color of purple, Fig. 6 right panel) to $PM_{2.5}$ concentrations. During warm periods, fine particle fraction is approximately 50% within the city and less than 40% in the villages close to Bucharest where the agricultural activities increase the $PM_{10}$ fraction (Fig. 6 left panel). The main rivers and lakes within Bucharest's perimeter are clearly sinks for small particles, producing lower fine mode fractions in both seasons."

**3.2.3 Evaluation of the model performance to resolve different types of environment**

**General comment:**

RefC: I see the relative differences seem to be mostly negative for the traffic datasets. To this extent I believe that the model is not suited to represent traffic congestions were multiple instances of vehicle stop/starts can result in higher NO2/Pm concentrations. I would also point out that vehicle types and ages, if not represented by the model, can also lead to these underestimations. Please update the discussions if this is the case.

AuthR: As we mentioned in the text (page 6 lines 176 - 182) we did not find official traffic data for Bucharest published for the past period. For this reason, the traffic values were estimated using data provided by the National Institute of Statistics and the geofabrik.de database. The use of estimated and not measured traffic data could be a cause for these underestimations. Nevertheless, the relative differences are less than 10%.

AuthCM: Page 14, line 294: added text "which can be also influenced by the low number of available fixed stations in each environment"

**Comment 1.**

RefC: Figure 3. Why are the standard deviations missing from the NO2 cold season traffic measurements?

AuthR: There was a bug in plot script. The figure was replaced with correct data.

AuthCM: Figure 3 was replaced.

**3.3 Mapping atmospheric pollution in Bucharest**

**Comment 1.**

RefC: Line 307: ...."Sinks related to the green areas and water bodies regions are identified in green colours..." Green as in all shade of green or as in a specific shade (e.g. dark green)?

AuthR: Dark green represents the water bodies and the forest regions. Other urban green regions are in various shades of green.

AuthCM: Page 14, line 307: "green colors" was changed to "dark green colors".

**Comment 2.**

RefC: There seems to be some inconsistency in the discussion at line 307 – 309 ... "overall NO2 concentration is higher during the warm period"... with lines 312 – 315 "At the level of the city of Bucharest, the average value of the NO2 concentration as estimated by the model for the warm season is 16.66 ± 4.04 ppb, while for the cold season it is 18.75 ± 1.98 ppb" (suggesting higher concentrations in the cold). Also, the upper panel of figure 4 seems to suggest that NO2 concentrations are indeed higher in the warm period.

AuthR: Thank you for pin pointing this inconsistency, it seems that $NO_2$ values were reversed. The correction was made in the text.

AuthCM: Page 15, lines 312-315: "At the level of the city of Bucharest, the average value of the $NO_2$ concentration as estimated by the model for the warm season is 16.66 ± 4.04 ppb, while for the cold season it is 18.75 ± 1.98 ppb" was changed to "At the level of the city of Bucharest, the average value of the $NO_2$ concentration as estimated by the model for the cold season is 16.66 ± 4.04 ppb, while for the warm season it is 18.75 ± 1.98 ppb"

**Comment 3.**

RefC: Line 316: "anthropocentric agglomerations"? Is it not anthropogenic?

AuthR: Modified according to the reviewer's note.

AuthCM: Page 15, line 316: "anthropocentric agglomerations" was changed to "anthropogenic".

**Comment 4.**

RefC: Line 322: The phrase "sources are more homogeneous" is not precise - it's not the sources that are homogeneous, but rather the distribution or concentration of PM10.

AuthR: Thank you for pin pointing the inconstancy, the text has been modified according to the reviewer's note.

AuthCM: Page 15, line 322: " sources are homogeneous" was changed to "sources distribution are homogeneous"

**Comment 5.**

RefC: Line 328: Please improve the clarity of the sentence, "Traffic sources are less effective..." Maybe "Traffic emissions have less impact" ... also consider improving the "reduce under reduced" end part of the sentence.

AuthR: The recommended corrections were done in the text.

AuthCM: Page 15, line 238: "Traffic sources are less effective during the cold season, when the chemical processes are reduced under reduced sunlight." was changed to "Traffic emissions have less impact during the cold season, as chemical processes diminish due to limited sunlight."

**Comment 6.**

RefC: Please update figure 5 with a large font size as seen in figures 3 and 4.

AuthR: The recommended corrections were done.

AuthCM: Figure 5 (now Figure 6): The fonts for scales were increased to be more legible.

**Conclusions**

**Comment 1.**

RefC: Lines 349-350. Consider replacing "correlated with" with "coupled with" to improve the clarity of the sentence.

AuthR: The recommended corrections were done.

AuthCM: Page 17, lines 349-350: "The slightly higher mean squared error values correlated with smaller ..." was changed to "The slightly higher mean squared error values coupled with smaller ..."

**Comment 2.**

RefC: Also, the sentence in lines 352-353 could benefit from minor grammatical improvements.

AuthR: Done, see changes below.

AuthCM: Page 17, lines 352-353: "Even if the route selected for the two mobile measurement campaigns included all urban structures, the limitation of car access remains a source of error, which can lead to an underestimation of the concentration of pollutants."

**Comment 3.**

RefC: Expanding on the ideas presented in sentences 355-356 would strengthen the article's overall argument. Consider adding a paragraph in the results section to further explore this topic.

AuthR: Modified according to the reviewer's note.

AuthCM: Page 14, line 196: added text "It should also be taken into account that, in the absence of spatio-temporal emission inventories with a high spatial resolution and traffic data, modeled data were used."

**Comment 4.**

RefC: The sentence at line 359 can also benefit from minor grammatical improvements.

AuthR: : Done, see changes below.

AuthCM: Page 17, line 359: "Fine particles dominate during the cold season, although they remain at high levels during the warm season as well."

**Comment 5.**

RefC: Line 365 – 367. Consider splitting the paragraph into 2 sentences to improve the clarity of the overall message.

AuthR: Modified according to the reviewer's note, see changes below.

AuthCM: Page 18, lines 365-367: "The approach presented in this paper can be adjusted for high-resolution mapping of $NO_2$, PMs and UFP in other cities as well. This is feasible as long as the urban structures are well-characterized and there is a fairly dense and diverse network of in situ monitoring stations, whose observational data can be used for model calibration and validation."

**Final comment**

RefC: I see the authors have identified the advantages of this approach with regards to adding policymakers in local administrations. To this extent I believe the study could benefit from one additional conclusion. Based on this novel information, how should the local municipality address the current distribution of air quality monitoring stations? Would the city benefit from additional AQ stations and/or any specific spatial configuration? Based on the model output, have the authors identified any inconsistencies with respect to the current land use configuration in Bucharest? And if so, how can the local administration address these possible issues?

AuthR: The objective of the paper is to set up the LUR model to ingest mobile measurements and to provide pollutants maps for the city of Bucharest. The focus of the paper is not the AQ stations locations or distribution, nor the land use configuration. Nevertheless the authors can provide several hypothesis, added in the Conclusion section.

AuthCM: Page 18, line 364: added text "Based on these findings, a more extended certified AQ station networks would be beneficial for human health related pollutants monitoring, as well as inclusion of fine particles measurements among these."

---

## Author Comment (AC2)

Dear referee,

Thank you very much for the valuable comments to our paper and suggestions for article improvements. The manuscript has been modified accordingly.

Below are the answers and modification on the manuscript. In the following: "RefC" is the comment from Referee, "AuthR" is the author's response and "AuthCM" represents the author's changes to the manuscript. Page and line number refer to the page and line number in the version submitted for discussion.

**Specific comments**

**Introduction**

**Comment 1.**

RefC: The introduction effectively outlines the problem and objectives but could benefit from a clearer emphasis on how this research fills existing gaps compared to other studies. Adding brief references to similar studies in other European cities could strengthen its relevance.

AuthR: Modified according to the reviewer's note.

AuthCM: Page 2, line 42: added references (Xu et al. (2021b); Knibbs et al. (2018); Lee et al. (2019)).

**Methodology**

**Comment 1.**

RefC: Explain the rationale behind the 3-point moving average for outlier removal. How does this choice affect the spatial resolution and data reliability?

AuthR: This choice does not affect the spatial resolution of the data, nor the data reliability, it just identifies the observation that differs significantly from its neighbors. The observation is invalidated when it is 1.5 times higher/lower than the average of the 3-points, observation included. In this way the spikes are removed and not induce additional biasis and the real concentrations (even if they are higher) are kept. The 3-points window was chosen due to the spatial relevance, the pollutants concentrations are similar in a given area, which is rapidly passed by the moving car. To be more clear to the reader the text has been adjusted.

AuthCM: Page 4, line 111: "3 data points window was used to remove data-points with values higher and lower than 1.5 times the window mean." was changed to "3 data points was used to invalidate the data-points with values higher and lower than 1.5 times the window mean."

**Results**

**Comment 1.**

RefC: The analysis in Figure 5 (PM2.5/PM10 ratio maps) needs refinement.

AuthR: Modified according to the reviewer's note, see changes to manuscript.

AuthCM: Page 16, lines 333-337: changed to "The model shows the fine particle fraction ($PM_{2.5}$ / $PM_{10}$) to be larger during the cold periods, compared to warm periods, with fine particles accounting for up to 95% of the $PM_{10}$ concentration (Fig. 6). This is explained by the fact that household activities generate predominantly small particles and higher percentages are seen in the peri-urban regions (outside of Bucharest) where the house heating are contributing more (lighter color of purple, Fig. 6 right panel) to $PM_{2.5}$ concentrations.

During warm periods, fine particle fraction is approximately 50% within the city and less than 40% in the villages close to Bucharest where the agricultural activities increase the $PM_{10}$ fraction (Fig. 6 left panel). The main rivers and lakes within Bucharest's perimeter are clearly sinks for small particles, producing lower fine mode fractions in both seasons.

**Comment 2.**

RefC: Borders in Tables: When zooming into the document, gaps are noticeable in the outside borders of tables. This may be due to uneven line weights, misaligned elements, or incomplete formatting. Please address this.

AuthR: Indeed, it was a technical issues in latex.

AuthCM: The Table borders have been redone.

**Discussion**

**Comment 1.**

RefC: Acknowledge the limitations of the mobile measurement route. For instance, areas with restricted car access might lead to underestimations.

AuthR: The configuration of the routes was designed to comprehensively cover the entire city while considering key factors: a) Pollutant Sources and Dispersion: Routes were planned to include areas with major emission sources and account for pollutant dispersion directions. b) Urban Typologies: The campaign ensured coverage of all relevant urban area types (e.g., residential, industrial, traffic-heavy zones) to meet the requirements of the dispersion model and provide representative data for the mixed-effects LUR model. This approach ensured that the campaign provided a robust dataset for high-resolution pollutant mapping while accounting for logistical, meteorological, and statistical considerations. Of course, the routes had to be designed to exclude the restricted areas for cars and in this way to exclude some urban areas (e.g. parks, agricultural zones and water bodies). To be more clearer to the reader the text has been adjusted.

AuthCM: Page 4, line 107: added text "The mobile measurement route is limited to the areas where the car has access, excluding some urban areas (e.g. parks, agricultural zones and water bodies)."

**Comment 2.**

RefC: The following suggestions would help the clarity:
Line 97: "...characterized by hot summers and cold winters." (clarify structure for conciseness).

AuthR: Modified according to the reviewer's note, see changes to manuscript.

AuthCM: Page 4 Lines 96-97: changed to " "Located in the southeastern part of Romania, in the Romanian Plain, Bucharest has a humid continental climate, characterized by hot summers , cold winters and two short transitional seasons, spring and autumn"

**Comment 3.**

RefC: Line 92-93: "...more industrialized, hosting a variety of manufacturing plants, such as machinery, textiles..." (remove redundancy).

AuthR: The recommended corrections were done in the text.

AuthCM: Page 4 lines 92-93: changed to "Most of the production sectors, such as machinery, textiles, chemicals, electronics, and business parks, all contributing significantly to the economic base of Bucharest, are located in the southern and western areas."

**Technical corrections**

**Comment 1.**

RefC: Line 164-165: aggregated values of the spatial predictor variables calculated in circular buffers with radii between 25 m and 2 km."
For more Clearer phrasing: "...aggregated values of spatial predictor variables calculated within circular buffers ranging from 25 m to 2 km in radius."

AuthR: The recommended corrections were done in the text.

AuthCM: Page 6 lines 164-165: changed to "...aggregated values of spatial predictor variables calculated within circular buffers ranging from 25 m to 2 km in radius."

**Comment 2.**

RefC: Line361: "...in pointed pollutant variability mostly during warm season and higher concentrations..."
Suggestion: "...pinpointed pollutant variability mostly during the warm season and higher concentrations..." Reason: "Pinpointed" should be one word, and "the" is required before "warm season."

AuthR: The recommended corrections were done in the text.

AuthCM: Page 18 line 361: changed to "...pinpointed pollutant variability mostly during the warm season and higher concentrations..."

---

## Author Comment (AC3)

Dear referee,

Thank you very much for the valuable comments to our paper and suggestions for article improvements. The manuscript has been modified accordingly.

Below are the answers and modification on the manuscript. In the following: "RefC" is the comment from Referee, "AuthR" is the author's response and "AuthCM" represents the author's changes to the manuscript. Page and line number refer to the page and line number in the version submitted for discussion.

**Specific comments**

**Introduction**

**Comment 1.**

RefC: Expand the explanation of the limitations of alternative models, such as dispersion models. For example, discuss the dependence of dispersion models on detailed meteorological data and high computational capacity, contrasting this with the simplicity and efficiency of LUR.

AuthR: Modified according to the reviewer's note,see changes to manuscript.

AuthCM: Page 2, lines 49-51: "(b) linear regression is one of the most used fine-scale spatial interpolation methods because it is fast, easy to implement, and does not require high computing power, and (c) a LUR model does not require detailed information on atmospheric conditions as input data ..." was changed to "(b) linear regression is one of the most used fine-scale spatial interpolation methods because it is fast, easy to implement (Hoek et al, 2008; Jerrett et al, 2005), and does not require high computing power such as computational fluid dynamics based on large-Eddy simulation or Reynolds-averaged Navier–Stokes approaches (Lin et al, 2023, 2024), and (c) a LUR model does not require detailed information on atmospheric conditions and an emission inventory as input data. LUR model usually requires measurement data and land-use predictor variables (e.g. CORINE dataset) ...."

**Comment 2.**

RefC: Develop a specific section addressing the short-term risks associated with high concentrations of PM10 and NO2. Include information about cardiovascular, respiratory, and even immune system impacts.

AuthR: We added several sentences related to the short-term human health risks associated with high PM and $NO_2$ concentrations.

AuthCM: Page 2, lines 21-23: "These compounds are often associated with the onset of multiple health issues including cardiovascular diseases, asthma or lung cancer" was changed to "Numerous epidemiological studies related short- and long-term $PM_{10}$ and $NO_2$ exposure with mortality and morbidity (). Short-term exposure to high concentrations of pollutants can be related to both minor discomfort, such as irritation of the eyes, respiratory tract, or skin, and serious conditions, such as asthma, pneumonia, bronchitis, chronic obstructive pulmonary disease and heart problems (Liu et al, 2022; Hasegawa et al, 2023). Furthermore, years of continuous exposure to PM were shown to be associated with both newborn mortality and cardiovascular disorders. A $PM_{2.5}$ concentration increase with 10 $\mu g/m^3$ was associated with a increase of 0.67% – 1.04% (Hamanaka and Gökhan 2018) in all-cause mortality, 0.52% in cardiovascular hospital admissions and 1.74% increase in respiratory admissions (Hasegawa et sl 2023). While, a $PM_{10}$ concentration increase with 10 $\mu g/m^3$ was associated with a 43% increase of fatal coronary heart disease (Hamanaka and Gökhan 2018) and 39.31% of deaths from cardiovascular diseases from short-term exposure (Seihei et al, 2024). A smaller impact is foreseen in the case of short-term exposure to $NO_2$ concentration, when an 10 ppb increase of concentration was associated with 0.19% increase in all-cause mortality in US (Hamanaka and Gökhan 2018).

**Comment 3.**

RefC: Are there comparative studies with other methods in cities similar to Bucharest that could enrich the justification presented?

AuthR: Unfortunately, there are no recent studies based on LUR methods or other methods of assessing air quality in metropolitan areas that can be compared with the metropolitan area of Bucharest, from the point of view of street network, traffic, urban and industrial development. In fact, this was one of the reasons why Bucharest was selected as a pilot station in the European RI-URBANS project. The mixed-effect LUR approach was already mentioned for several cities.

AuthCM: none

**Methodology**

**Comment 1.**

RefC: Add detailed maps of the study area, highlighting industrial, residential, and commercial zones. Include the routes of mobile measurements and collection points to contextualize the spatial distribution.

AuthR: Modified according to the reviewer's note.

AuthCM: A map with diverse land use types present in the Bucharest metropolitan area and the routes from the measurement campaign has been added as Figure 1.

**Comment 2. Part A**

RefC: Explain the criteria for buffer size selection and the reasons for using varying sizes.

AuthR: Buffer analysis is a common technique in GIS. The sizes of the buffers used in this study are those established within the European Study of Cohorts for Air Pollution Effects (ESCAPE) for the development of LUR models. Variable buffer sizes are applied to create buffers around raster datasets used for analyzing spatial relationships between continuous surfaces, such as identifying areas of land use that are a certain distance from a street segment.

AuthCM: Page 7, line 207: added text "The sizes of the buffers are those established within ESCAPE project for the development of LUR models. These buffer sizes are used to determine the spatial proximity of different features by defining a distance zone around the features."

**Comment 2. Part B**

RefC: Justify the use of the moving average filter, considering its role in removing outliers and enhancing model accuracy.

AuthR: The models are obtained by fitting the LUR model structures to the observed input-output data. If the data obtained from the measurements contain "outliers" or "outlier data points", they can negatively influence the model results. The moving average is the most common tool used to enhance the accuracy of the model by smoothing out data fluctuations caused by random variations or noise. By calculating the average of a set of time-series data, moving averages can reveal underlying trends and patterns that would be difficult to see otherwise.

AuthCM: none

**Comment 3.**

RefC: Include a comparative table presenting the advantages and disadvantages of alternative methodologies, such as satellite-based models versus hybrid models like LUR/mixed-effects.

AuthR: Of course, there are many methodologies used for air quality. In our article, we mention the methodologies close to our approach.

AuthCM: none

**Comment 4.**

RefC: Were sensitivity tests conducted to evaluate the impacts of different combinations of predictive variables?

AuthR: Assessing the impact of different combinations of predictive variables is the subject of another article and was not included in this study. In this paper, only the models for which the highest values of $R^2$ were obtained are presented and discussed. (lines 210-211: "Further, only the results and performances of the LUR models for which the highest adjusted $R^2$ value was obtained are discussed")

AuthCM: none

**Comment 5.**

RefC: How did varying buffer sizes influence the results, and was cross-validation performed to determine the optimal parameters?

AuthR: The results of the learning process were influenced by the variation of the size of the buffers by obtaining different values for the statistical parameters $R^2$, variance inflation factor (VIF) and p_value, the parameters that were used to select the predictor variables used in the LUR models. After applying the selection criterion of the best models (Section 3.1.2 "Predictor variable selection"), a cross-validation was performed for all selected models. The models that obtained the highest $R^2$ score were considered to have the optimal parameters.

AuthCM: none

**3 Results and discussions**

**Comment 1.**

RefC: Relocate the methodological descriptions from sections 3.1, 3.1.1, 3.1.2, and 3.2 to the methodology section, facilitating a more focused discussion of the results.

AuthR: Due to the complexity of the process of cross-validation of the model with observational data, we would prefer that the description of this process and the results obtained be presented in section 3 "Results and discussions".

AuthCM: The methodological descriptions of Sections 3.1, 3.1.1, 3.1.2 were relocated to the methodology section.

**Comment 2.**

RefC: It would be beneficial to explain in detail why the model underestimates PM10 levels in urban areas. Including maps illustrating the spatial distribution of pollutants in different environments would enhance the section on environmental types.

AuthR: The 2018 CORINE land cover data used in the model is currently the latest iteration. For this reason, the model could underestimate the level of PM10 in urban areas, as it does not fully cover the changes made

in the last 6 years in the metropolitan area of Bucharest. Nevertheless, the relative differences are around 10%. However, the results obtained from the model agree with the measured data. We considered that the maps presented in figures 2 and 4 are relevant to illustrate the spatial distribution of pollutants in different environments.

AuthCM: A map with diverse land use types present in the Bucharest metropolitan area and the routes from the measurement campaign has been added as Figure 1.

Page 14, line 194: Added: "which can be also influenced by the low number of available fixed stations in each environment"

**Comment 3.**

RefC: What were the main technical challenges in modeling industrial and high-traffic areas?

AuthR: The lack of data, especially traffic data, were the main technical challenges.

AuthCM: none

**Comment 4.**

RefC: Add graphs showing the differences between predictions and measured values, highlighting seasonal variations.

AuthR: We consider that Figure 2 and Figure 3 from the submitted manuscript are relevant both to emphasize the differences between the modeled data and the data measured at the fixed stations as well as the seasonal variation.

AuthCM: none

**Conclusions**

**Comment 1.**

RefC: Detail how citizen involvement could improve data collection, including examples of using bicycles or pedestrians to access restricted areas.

AuthR: OK, see changes to manuscript.

AuthCM: Page 17, line 355: added text "Data sets systematically collected by citizens during daily (repeated) activities or walks could provide improved estimates of spatial variability for these areas (Snyder et al., 2013; Hankey and Marshall, 2015; Van den Bossche et al., 2016 and 2018). The citizen involvement increases the pollutants data collection on areas restricted for cars or bicycles and enable the possibility to study the sinks on green or water areas, but are based only on low cost sensors.

**Comment 2.**

RefC: Discuss how the methods could be adjusted for cities with similar urban characteristics, detailing the data requirements and necessary adjustments.

AuthR: The usually required input data in LUR are measurement data and land-use predictor variables. Depending on the purpose for which the model is used, data population density (available from national or European statistics catalogues) and traffic intensity variables (available from the national statistics, or modeled) can also be used as input data. Measured data can be obtained from the national air quality monitoring network, if there are enough stations with a distribution to cover the entire area of interest or, as in the case of

the study presented in this manuscript, from the campaigns. The LUR models must be fitted to the measurements specific to the area of interest to obtain the regression coefficients used to estimate the concentrations.

AuthCM: Page 18, line 366: added text "...in other cities as well, using the series of predictor variable identified in this study as necessary. This is feasible ..."

---

## Author Comment (AC4)

Dear editor,

Here are some technical and minor language corrections from the authors. In the following "AuthCM" represents the author's changes to the manuscript. Page and line number refer to the page and line number in the version submitted for discussion.

**Technical corrections**

**Comment 1.**

AuthCM: Page 9, lines 230-231 The RMSE values were corrected. In the original document, the MSE (mean squared error) was listed as the RMSE value."

**Comment 2.**

AuthCM: Page 18, lines 368-369: Corrected the surname of the main author "Talinau" to "Talianu"

---

## Referee Report (RR1)

The revised manuscript demonstrates substantial improvements in methodological clarity, contextualization, and presentation, addressing the majority of concerns raised during initial review. The authors have successfully strengthened the introduction, expanded methodological justifications, and enhanced the discussion of practical implications for policymakers. While the bulk of the revisions is now complete, minor technical corrections are required to ensure consistency, precision, and reproducibility. Specifically, attention should be given to: clarifying the spatial extent depicted in Figure 1; ensuring consistent map delineation with the described study area; resolving the discrepancy in $R^2$ values between lines 160-163 and 247-252; and incorporating details about the in-situ instrument(s) at the MARS facility in relation to lines 279-280.

**Technical comments**

Figure 1.

The spatial extent of Bucharest considered in the land use regression requires clarification (e.g., whether it aligns with the map boundary or a narrower study area). A precise delineation of this extent would enhance methodological transparency and allow readers to evaluate the land use classes integrated into the model. Additionally, the absence of standard cartographic elements—such as a scale, orientation, or geographic coordinates (e.g., latitude/longitude, as provided in Figure 2)—limits the interpretability of the spatial data.

Lines 133 – 135 *…. window on 3 data points window…. values higher and lower than*

*1.5 times the window mean.*

The clarity of the text can be improved:  A moving average filter with a 3-data-point window was used to remove data points with values exceeding 1.5 times the window mean, above or below.

Line 199 – 200 *... over an area of approximately 240 km$^2$ (the entire area of the city*

*of Bucharest).*

Going back to the discussion on figure 1. This area should be clearly delineated on the map.

Lines 160 – 163: *The performance of the model has been evaluated in three steps. First, a subset containing 15% of data collected through mobile measurements (and not used to tune the model) was used for cross-validation. This percentage represents the optimal value for which the models developed in this study can recognize the*

*relationship between the attributes of the input data and the output variable with $R^2$ score greater than 0.75.*

Lines 247 – 252: *The performance of each model (one for each type of pollutant) was tested in three steps. First, the outputs of the model have been cross-validated against mobile measurements on the route (the 15% randomly selected mobile datasets not used for training). The percentage of 15% kept for testing represents the optimal value for which the models developed in this study can recognize the relationship between the attributes of the input data and the output variable with $R^2$ score greater than 0.5.*

Is it greater than 0.5 or greater than 0.75? To ensure clarity and eliminate redundancy, please remove one of the duplicated paragraphs and clarifying the R2 score.

Line 279 – 280. *More information about what type of variables are measured by*

*NAQMN is given in (Ilie et al, 2023).*

An additional sentence should be added regarding the in-situ instrument(s) at the Magurele Center for Atmosphere and Radiation Studies (MARS).

---

## Author Response (AR2)

*Dear referee,*

*Thank you very much for the valuable comments to our paper and suggestions for article improvements. The manuscript has been modified accordingly.*

*Below are the answers and modification on the manuscript. In the following: "RefC" is the comment from Referee, "AuthR" is the author's response and "AuthCM" represents the author's changes to the manuscript.*

RefC: "The revised manuscript demonstrates substantial improvements in methodological clarity, contextualization, and presentation, addressing the majority of concerns raised during initial review. The authors have successfully strengthened the introduction, expanded methodological justifications, and enhanced the discussion of practical implications for policymakers. While the bulk of the revisions is now complete, minor technical corrections are required to ensure consistency, precision, and reproducibility. Specifically, attention should be given to: clarifying the spatial extent depicted in Figure 1; ensuring consistent map delineation with the described study area; resolving the discrepancy in R2 values between lines 160-163 and 247- 252; and incorporating details about the in-situ instrument(s) at the MARS facility in relation to lines 279-280.

**Technical comments**

Figure 1.

The spatial extent of Bucharest considered in the land use regression requires clarification (e.g., whether it aligns with the map boundary or a narrower study area). A precise delineation of this extent would enhance methodological transparency and allow readers to evaluate the land use classes integrated into the model. Additionally, the absence of standard cartographic elements—such as a scale, orientation, or geographic coordinates (e.g., latitude/longitude, as provided in Figure 2)—limits the interpretability of the spatial data. "

AuthR: The recommended corrections were done.

AuthCM: Figure 1 was replaced, added the geographic coordinates, orientation, scale and a rectangular for area used in the model.

RefC: "Lines 133 – 135 …. *window on 3 data points window…. values higher and lower than 1.5 times the window mean*.

The clarity of the text can be improved: A moving average filter with a 3-data-point window was used to remove data points with values exceeding 1.5 times the window mean, above or below."

AuthR: The recommended corrections were done in the text.

AuthCM: "A moving average filter with a 3-data-point window was used to remove data points with values exceeding 1.5 times the window mean, above or below."

RefC: "Line 199 – 200 ... over an area of approximately 240 km2 (the entire area of the city of Bucharest). Going back to the discussion on figure 1. This area should be clearly delineated on the map."

AuthR: The recommended corrections were done.

AuthCM: Figure 1 was replaced, added a rectangular pin pointing the area used in the model. Also it was added in the figure caption "dotted rectangular represent the modelled area"

*RefC: "Lines 160 – 163: The performance of the model has been evaluated in three steps. First, a subset containing 15% of data collected through mobile measurements (and not used to tune the model) was used for cross-validation. This percentage represents the optimal value for which the models developed in this study can recognize the relationship between the attributes of the input data and the output variable with $R^2$ score greater than 0.75.*

*Lines 247 – 252: The performance of each model (one for each type of pollutant) was tested in three steps. First, the outputs of the model have been cross-validated against mobile measurements on the route (the 15% randomly selected mobile datasets not used for training). The percentage of 15% kept for testing represents the optimal value for which the models developed in this study can recognize the relationship between the attributes of the input data and the output variable with $R^2$ score greater than 0.5.*

*Is it greater than 0.5 or greater than 0.75? To ensure clarity and eliminate redundancy, please remove one of the duplicated paragraphs and clarifying the R2 score."*

AuthR: Thank you for pointing out the redundancy and typo in the second paragraph.

AuthCM: We deleted most of the second appearance (lines 247-252) in the text and rephrased as follows: "The performance of each model (one for each type of pollutant) was tested following three steps described in details in section 2.3. Moreover, in the first step, the 15% of data kept for testing covers all possible situations ………"

*RefC: "Line 279 – 280. More information about what type of variables are measured by NAQMN is given in (Ilie et al, 2023).*

An additional sentence should be added regarding the in-situ instrument(s) at the Magurele Center for Atmosphere and Radiation Studies (MARS)."

AuthR: A sentence related to MARS site and measurements was added.

AuthCM: It was added the sentence "Detailed description of MARS site is given in (Pirloaga et al., 2023), where continuous PM concentrations are performed using optical particle counters (Marmureanu et al.,2019) and gases analysers (Castell et al., 2018)."

and added the following to the reference section:

"Mărmureanu, L.; Marin, C.A.; Andrei, S.; Antonescu, B.; Ene, D.; Boldeanu, M.; Vasilescu, J.; Viţelaru, C.; Cadar, O.; Levei, E. Orange Snow—A Saharan Dust Intrusion over Romania During Winter Conditions. Remote Sens. 2019, 11, 2466. https://doi.org/10.3390/rs11212466"

"Pîrloagă, R.; Adam, M.; Antonescu, B.; Andrei, S.; Ştefan, S. Ground-Based Measurements of Wind and Turbulence at Bucharest–Măgurele: First Results. Remote Sens. 2023, 15, 1514. https://doi.org/10.3390/rs15061514"

"Castell, N.; Schneider, P.; Grossberndt, S.; Fredriksen, M.F.; Sousa-Santos, G.; Vogt, M.; Bartonova, A. Localized real-time information on outdoor air quality at kindergartens in Oslo, Norway using low-cost sensor nodes, Environmental Research, Volume 165, 2018, Pages 410-419, ISSN 0013-9351, https://doi.org/10.1016/j.envres.2017.10.019"